# DiffTrans: Differentiable Geometry-Materials Decomposition for Reconstructing Transparent Objects

**Changpu Li**[*1], **Shuang Wu**[*3], **Songlin Tang**[1], **Guangming Lu**[1], **Jun Yu**[1], **Wenjie Pei**[†1,2]

[1]Harbin Institute of Technology, Shenzhen
[2]Peng Cheng Laboratory
[3]Nanjing University
{lichangpu29, wushuang9811, wenjiecoder}@outlook.com;
tangsonglin@stu.hit.edu.cn;
{luguangm, yujun}@hit.edu.cn

## Abstract

Reconstructing transparent objects from a set of multi-view images is a challenging task due to the complicated nature and indeterminate behavior of light propagation. Typical methods are primarily tailored to specific scenarios, such as objects following a uniform topology, exhibiting ideal transparency and surface specular reflections, or with only surface materials, which substantially constrains their practical applicability in real-world settings. In this work, we propose a differentiable rendering framework for transparent objects, dubbed *DiffTrans*, which allows for efficient decomposition and reconstruction of the geometry and materials of transparent objects, thereby reconstructing transparent objects accurately in intricate scenes with diverse topology and complex texture. Specifically, we first utilize FlexiCubes with dilation and smoothness regularization as the iso-surface representation to reconstruct an initial geometry efficiently from the multi-view object silhouette. Meanwhile, we employ the environment light radiance field to recover the environment of the scene. Then we devise a recursive differentiable ray tracer to further optimize the geometry, index of refraction and absorption rate simultaneously in a unified and end-to-end manner, leading to high-quality reconstruction of transparent objects in intricate scenes. A prominent advantage of the designed ray tracer is that it can be implemented in CUDA, enabling a significantly reduced computational cost. Extensive experiments on multiple benchmarks demonstrate the superior reconstruction performance of our *DiffTrans* compared with other methods, especially in intricate scenes involving transparent objects with diverse topology and complex texture. The code is available here.

## 1 Introduction

The reconstruction of geometry and materials for transparent objects poses a highly complex and ill-posed problem (Kutulakos & Steger, 2008). This is primarily due to the intricate interplay between light refraction among an object's surfaces and its surrounding environment. As a result, compared to opaque objects, the appearance of transparent objects is more intricately entangled with their geometry, such that even a slight change in scene parameters can lead to significant variations in their appearance.

Early approaches employ environment matting (Zongker et al., 1999; Matusik et al., 2002) to reconstruct transparent objects, which could exploit specialized hardware setups (Matusik et al., 2002). Recently, differentiable rendering have demonstrated significant success in novel view synthesis and inverse rendering, unlocking new potential for reconstructing transparent objects. One typical way is to leverage the eikonal field (Bemana et al., 2022; Deng et al., 2024) to model the march-

---

[†]Corresponding author.
[*]Equal contribution.

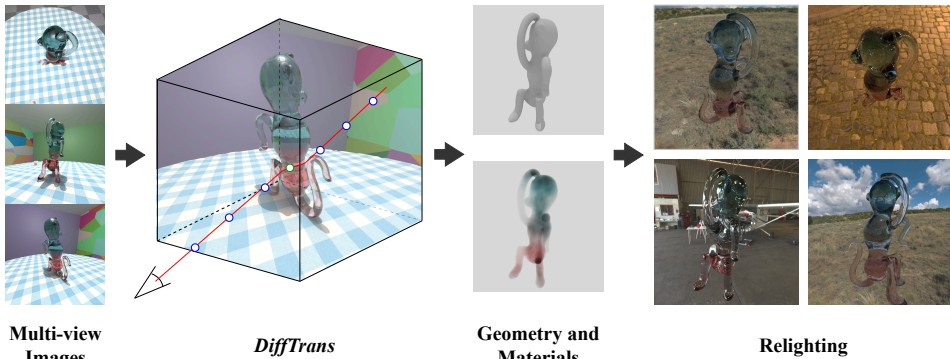

| Multi-view Images | *DiffTrans* | Geometry and Materials | Relighting |

Figure 1: Overview of the proposed *DiffTrans*. Our approach reconstructs the geometry and materials of the transparent objects with diverse topology and complex texture from a set of multi-view images using differentiable mesh ray tracer, enabling scene editing capabilities such as relighting.

ing process of light. Despite achieving promising fidelity in novel views and scenes with complex object composition, these methods struggle to extract reliable meshes due to the lack of constraints on surface geometry, thus unable to deal with objects with intricate meshes (Wang et al., 2023). Another representative way is based on surface reconstruction using representations such as neural SDF field (Yariv et al., 2020; Wang et al., 2021), triangle mesh (Lyu et al., 2020) or 3D gaussian splatting (Huang et al., 2025). However, these methods fail to reconstruct the geometry of transparent objects with complex textures. NEMTO (Wang et al., 2023), NeTO (Li et al., 2023) and NeRRF (Chen et al., 2023) neglect the materials of transparent objects, while Nu-NeRF (Sun et al., 2024) and TransparentGS (Huang et al., 2025) only model the surface materials of transparent objects. None of the aforementioned methods can effectively model transparent objects with complex internal absorptive texture, which are commonly observed in real-world scenes, such as jewels, glass decorations, and handmade resin.

In this paper, we propose to optimize the geometry and materials of the transparent objects simultaneously to improve the reconstruction quality. As shown in Figure 1, we devise a novel differentiable rendering framework, dubbed *DiffTrans*, which decouples the geometry and materials of the transparent objects with intricate topology and internal texture from a set of multi-view images, enabling scene editing capabilities such as relighting. Our *DiffTrans* consists of three stages that progressively facilitate the reconstruction of the geometry, environment and materials. First, we adopt FlexiCubes (Shen et al., 2023) as the iso-surface representation to recover the topology of the transparent objects from multi-view object silhouette. By incorporating dilation and smoothness regularization, we can efficiently obtain an initial geometry. Meanwhile, we employ a radiance field (Reiser et al., 2023) with a voxel grid and a triplane to recover the environment of the scene with out-of-mask regions of the input images. Finally, as the most crucial step, we design a recursive differentiable mesh ray tracer to further optimize and refine the geometry, index of refraction and absorption rate of the transparent objects in a unified and end-to-end manner. In particular, we implement the ray tracer in OptiX and CUDA, significantly enhancing computational efficiency.

In summary, we make following contributions:

- We design a novel differentiable rendering framework, dubbed *DiffTrans*, which allows for efficient decomposition and reconstruction of geometry and materials of the transparent objects, thereby reconstructing transparent objects accurately in intricate scenes with diverse topology and complex texture.

- We propose to employ FlexiCubes with dilation and smoothness regularization as the iso-surface representation, to reconstruct the initial geometry of transparent objects using only multi-view object silhouette. Concurrently, we can efficiently recover the environment of the scene using a environment light radiance field.

- We devise a differentiable recursive mesh ray tracer for the joint optimization of geometry, index of refraction and absorption rate of the transparent objects. The designed ray tracer is implemented in OptiX and CUDA to enhance optimization efficiency.

- Extensive experiments on both synthetic and real-world datasets demonstrate that our *Diff-Trans* achieves favorable reconstruction quality compared to other state-of-the-art methods for the transparent objects with different topology and intricate texture.

## 2 RELATED WORKS

**Transparent Object Reconstruction.** The methods fall into two categories: eikonal-based reconstruction and surface-based reconstruction. (Bemana et al., 2022) uses a eikonal field in a user-defined area to store IoR and solve the eikonal equation to get differentiable light paths. (Deng et al., 2024) stores the ray marching length and direction and impose straight ray restriction. (Lyu et al., 2020; Wu et al., 2018) leverages patterned background for extra ray supervision and optimize on a mesh. (Li et al., 2023) uses implicit surface and discard unreliable rays to improved the quality. (XU et al., 2022) further uses a hybrid surface representation of mesh and neural network to stabilize training. (Wang et al., 2023) combines implicit surface and environment matting to recover visually appealing result. (Chen et al., 2023) developed a mixture of DMTet (Shen et al., 2021) and neural network to achieve a robust mask-only training procedure. (Sun et al., 2024) proposes a ray-traced iterative strategy to reconstruct nested transparent objects. (Huang et al., 2025) adopts 3D gaussian splatting and light probe to promote the delicacy of reconstruction. (Wu et al., 2025) applies multi-level level sets and regularizations on pre-trained NeRF to adapt to translucent or thin objects. However, None of the aforementioned methods can handle transparent objects with complex internal absorptive texture.

**Inverse Rendering.** Inverse rendering aims to decompose visual input into geometry, material, and lighting. (Zhang et al., 2021) pioneers this direction by factorizing shape and spatially-varying reflectance from a pre-trained NeRF, utilizing surface normals and visibility approximations. To improve optimization efficiency, (Jin et al., 2023) leverages a tensorial representation to jointly reconstruct geometry and material properties under unknown illumination. Recently, explicit representations have been adapted for relighting tasks; (Gao et al., 2024) integrates BRDF decomposition and ray tracing into 3D Gaussian Splatting to achieve realistic point-based relighting. Addressing the complexities of global illumination, (Dai et al., 2025) introduces a multi-bounce path tracing framework combined with reservoir sampling to accurately recover scene parameters.

**Mesh-based Optimization.** Many differentiable renderers (Laine et al., 2020; Loper & Black, 2014; Jakob et al., 2022) support mesh representation. (Nicolet et al., 2021) introduces a mesh optimizer using the laplace matrix of mesh to avoid self-intersection and other problems during training. Besides directly optimizing on a fixed-topology mesh, some methods (Shen et al., 2021; 2023) take meshes as the proxy of other geometry representations, which are free from common artifacts during mesh optimization and greatly extended its application such as universal inverse rendering (Munkberg et al., 2022; Hasselgren et al., 2022). To overcome the inefficiency of optimizing through visibility gradient, (Mehta et al., 2023) derived a stronger supervision on mask error of meshes which is able to dynamically adapt to different topology. (Son et al., 2024) proposes a differentiable mesh representation that supports topology changes during the inverse process, while (Binninger et al., 2025) introduces an on-the-fly Delaunay tetrahedral grid strategy to ensure stable isosurface extraction for gradient-based optimization.

**NeRF-based Rendering.** NeRF (Mildenhall et al., 2020) achieves photo-realistic rendering by learning implicit scene representation with pure MLPs which incurs slow training speed. To this end, recent works seek to efficient explicit representation to avoid using deep MLPs. Plenoxels (Fridovich-Keil et al., 2022) uses pure explicit sparse voxel grid. DVGO (Sun et al., 2022) uses dense voxel grid, TensoRF (Chen et al., 2022) uses decomposed tensorial voxel grid, Instant-NGP (Müller et al., 2022) uses multi-resolution hash voxel grid and MERF (Reiser et al., 2023) uses low resolution dense voxel grid and high resolution tri-planes, these methods all employ tiny MLPs as voxel feature decoder to further improve representation capacity. Although these methods achieve great quality of novel view synthesis, they usually suffer from poor extracted geometry due to lack of geometry restriction in volume rendering, especially for transparent objects. The assumption of straight ray of volume rendering also ignores the refraction of light which happen frequently in transparent objects.

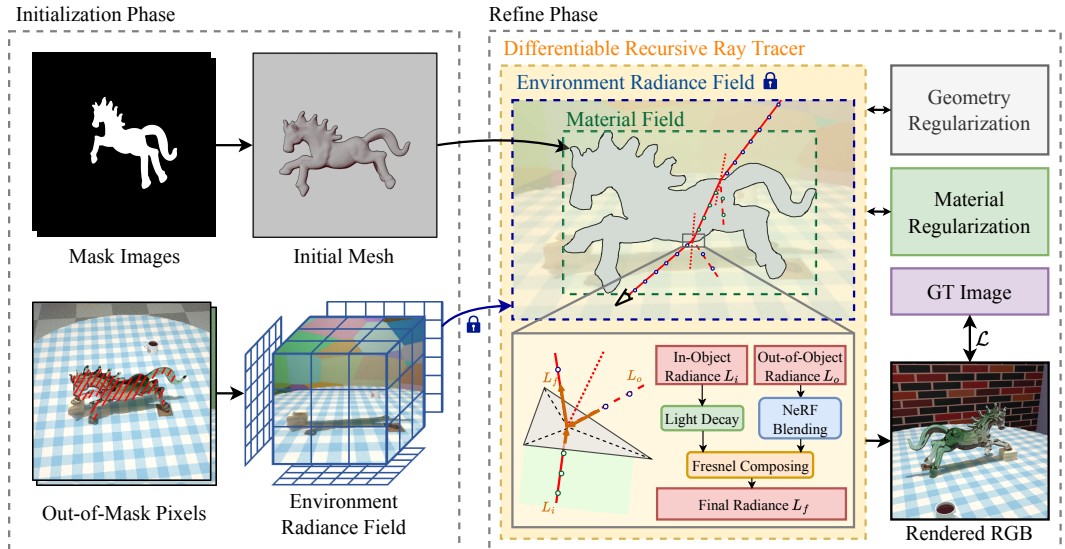

Figure 2: The framework of our *DiffTrans*. We commence by reconstructing the initial geometry from multi-view mask images, and recovering the environment light radiance field by employing pixels out of the mask regions (Section 3.2). In the refine phase, we first define the light decay within absorptive medium, and the Fresnel term used for blending in-object radiance and out-of-object radiance (Section 3.3). Then we optimize the geometry, IoR and absorption rate of the transparent objects simultaneously via our differentiable recursive mesh ray tracer, with geometry and material regularization (Section 3.4).

# 3 METHOD

## 3.1 OVERVIEW

Our *DiffTrans* aims to reconstruct the geometry and materials of transparent objects from a collection of multi-view images, the masks of the transparent objects in the scene and the corresponding camera parameters. Previous works, such as NeRRF (Chen et al., 2023), are limited to reconstructing transparent objects that exhibit ideal transmission and surface specular reflection, while Nu-NeRF (Sun et al., 2024) models only the surface materials, neglecting the internal absorption rate of transparent objects, which leads to subpar reconstruction quality when the textures of transparent objects are highly complex. In contrast, our *DiffTrans* employs the differentiable decomposition of both geometry and materials, and explicitly models the absorption rate of transparent objects, significantly enhancing the reconstruction quality. Figure 2 illustrates the overall framework of our *DiffTrans*. Similar to previous methods, we adopt a progressive training strategy to achieve more stable results. Specifically, we first utilize the multi-view masks of transparent objects along with the corresponding camera parameters to reconstruct the coarse geometry through differentiable rasterizing (Laine et al., 2020). Meanwhile, we employ the pixels in the out-of-mask regions of the multi-view images to recover the 3D environment. Subsequently, we introduce a novel recursive ray tracer to simultaneously optimize the geometry and materials of the transparent objects, including index of refractive (IoR) and absorption rate, in a unified and end-to-end manner.

**Assumptions.** The inherent ill-posed nature of the reconstruction for transparent objects necessitates simplifications in the scene. To address this, we make three key assumptions. Firstly, we postulate that each point within the transparent object has a consistent refractive index, which allows rays to propagate linearly inside the object, thereby eliminating the need for eikonal rendering. Secondly, we posit that the materials of the transparent objects solely consists of the absorption rate and index of refraction. This assumption is essential due to the substantial ambiguity in radiance caused by in-scattering and background when precise recovery of the geometry and background can not be achieved. Our final assumption is that the surface of the transparent object exhibits specular behavior, as inverse rendering of rough transparent objects remains a highly challenging problem that introduces excessive complexity. Although the scene has been simplified through these assumptions,

our *DiffTrans* framework is sufficiently capable of representing a majority of transparent objects in the real world.

## 3.2 GEOMETRY AND ENVIRONMENT INITIALIZATION

Different from reflection-only surfaces (Liu et al., 2023), the appearance of transparent objects is highly sensitive to the environment. In order to ensure the convergence of the optimization, we first initialize the geometry of the transparent objects and the environment light radiance field, providing a fast and reliable starting point for the subsequent optimization.

**Mask-based Geometry Initialization.** We leverage a differentiable rasterizer (Laine et al., 2020) to recover the initial geometry of transparent objects from multi-view object masks. Specifically, we employ FlexiCubes (Shen et al., 2023) as the isosurface representation to project the 3D mesh to the 2D image plane, and utilize the ground truth mask $\mathcal{M}_i$ to supervise the rendered images $\hat{\mathcal{M}}_i$ with L1-loss $\mathcal{L}_{\text{geo-init}}$. We observe that using only masks for supervision leads to numerous artifacts and cracks in the reconstructed mesh. To mitigate this issue, we apply dilation regularization to penalize the SDF values:

$$L_{\text{dilation}} = \frac{1}{N} \sum_i \text{sdf}(x_i), \tag{1}$$

where $\text{sdf}(x_i)$ is the SDF value at point $x_i$ randomly drawn from the canonical space. To mitigate potential noises during training, we apply smoothness regularization $\mathcal{L}_{\text{smooth}}$ to the depth $d$ and normal $\mathbf{n}$ of the objects in the screen space by penalizing the first and second-order gradients. Additionally, we adopt various other regularization and optimization strategies to enhance the quality of the initialized geometry, with further details provided in Section A.1 of the appendix.

**Environment Initialization.** We employ a radiance field (Reiser et al., 2023) with a coarse dense grid and fine triplanes, along with two iNGP-style (Müller et al., 2022) proposal grids to represent the environment of the scene. The environment light radiance field is initialized leveraging out-of-mask pixels in a NeRF-like manner:

$$\mathcal{L}_{\text{env-init}} = \sum_{i=0}^{N-1} \left\| \left( \hat{\mathcal{I}}_i - \mathcal{I}_i \right) \circ (1 - \mathcal{M}_i) \right\|_1, \tag{2}$$

where $\circ$ denotes element-wise product.

## 3.3 LIGHT INTERACTION WITH TRANSPARENT OBJECTS

Based on the initial geometry and environment obtained in the previous stage, we propose a novel differentiable recursive ray tracing to further optimize the geometry and materials of the transparent objects in a unified and end-to-end manner. Unlike previous works (Li et al., 2020) which utilize feed-forward approach to recover the transparent objects, our method reconstructs these objects in an analysis-by-synthesis manner that adheres to a series of physical laws, ensuring the reality of reconstruction. We will detail our ray tracer in Section 3.4.

**Light Interaction with Transparent Surfaces.** Since we do not consider the roughness of the transparent objects, the light interaction can be simplified to a deterministic process that does not require random sampling. We categorize the interaction between rays and the surface of the transparent objects into two types: reflection and refraction. The reflected light direction $\omega_r$ and refracted light direction $\omega_t$ are computed as follows by separating the outgoing direction into the components parallel and perpendicular to the surface normal:

$$\omega_r = 2 \left( \omega_i \cdot \mathbf{n} \right) \mathbf{n} - \omega_i, \tag{3}$$

$$\omega_t = \omega_t^\perp + \omega_t^\parallel, \tag{4}$$

$$\omega_t^\parallel = -\mathbf{n} \cdot \frac{1}{\eta} \sqrt{\eta^2 - \sin^2 \theta_i}, \tag{5}$$

$$\omega_t^\perp = -\frac{1}{\eta} \left( \omega_i - (\omega_i \cdot \mathbf{n}) \mathbf{n} \right), \tag{6}$$

where $\mathbf{n}$ denotes the normal direction, $\eta$ is the ratio of the IoRs between the outgoing medium and the ingoing medium, which is uniform per object, and $\theta_i = \omega_i \cdot \mathbf{n}$. When total internal reflection occurs, we consider only reflection and disregard refraction, i.e. $\eta^2 - \sin^2 \theta_i < 0$. The reflection rate $R$ and the refraction rate $T$ are calculated based on the Fresnel Equation:

$$R = \frac{1}{2}[(\frac{\eta_i \cos\theta_i - \eta_t \cos\theta_t}{\eta_i \cos\theta_i + \eta_t \cos\theta_t})^2 + (\frac{\eta_i \cos\theta_t - \eta_t \cos\theta_i}{\eta_i \cos\theta_t + \eta_t \cos\theta_i})^2], \tag{7}$$

$$T = 1 - R. \tag{8}$$

**Optical Transportation in Absorptive Medium.** According to our assumption in Section 3.1, we simplify the Radiative Transport Equation (Kajiya & Von Herzen, 1984) to the following form:

$$\frac{\partial}{\partial \omega} L(\mathbf{x}, \omega_i) = -\mu_t(\mathbf{x}) L(\mathbf{x}, \omega_i), \tag{9}$$

where $\omega$ denotes the propagation direction of the light, $L(\mathbf{x}, \omega_i)$ is the radiance at point $\mathbf{x}$ in direction $\omega_i$, and $\mu_t(\mathbf{x})$ is the absorption rate at point $\mathbf{x}$. The equation shows the decay of the radiance along the ray direction in the medium. The integral form of the equation is:

$$L(\mathbf{x}, \omega) = L(\mathbf{x}_0, \omega) \exp\left(-\int_{\mathbf{x}_0}^{\mathbf{x}} \mu_t(\mathbf{x}) d\mathbf{x}\right), \tag{10}$$

where $\mathbf{x}_0$ is the starting point of the ray. We calculate $L(\mathbf{x}, \omega)$ in a NeRF-like manner:

$$L(\mathbf{x}, \omega) = L(\mathbf{x}_0, \omega) \exp\left(-\sum_i \mu_t(\mathbf{x}_i)\Delta\mathbf{x}_i\right). \tag{11}$$

The absorption rate $\mu_t(\mathbf{x})$ is represented as a differentiable 3D texture (Müller et al., 2022) and optimized using our proposed differentiable mesh ray tracer.

## 3.4 DIFFERENTIABLE MESH RAY TRACER

Our differentiable mesh ray tracer simulates the light interactions with the transparent objects in a recursive manner, enabling the unified and end-to-end optimization of geometry, index of refraction and absorption rate. The recursive ray tracer is implemented using OptiX and CUDA, significantly reducing computational overhead.

**Recursive Ray Tracing.** We render the color of each camera ray by recursively tracing it until it no longer intersects with the transparent objects. Benefiting from our assumptions of coherent IoR and the absence of roughness, ray tracing can be simplified to a deterministic process without the need to estimate rendering integrals. We partition the sampling region into two parts: the interior and exterior of the object, sampling the absorption rate field and the environment field, respectively. To be specific, we process the rays in the following order:

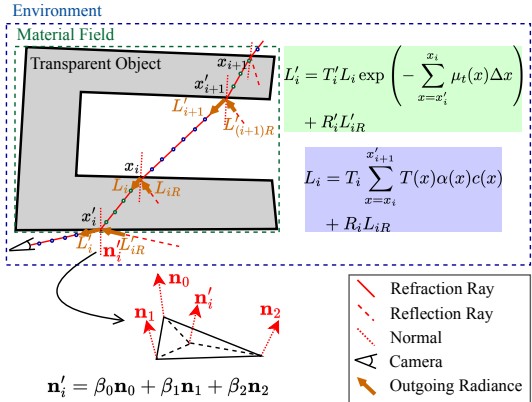

Figure 3: The recursive rendering process demonstrated with a light path. $L'_{iR}$ and $L_{iR}$ are the radiance of the reflected ray, which is abbreviated as a single variable for simplicity. $T_i$, $R_i$, $T'_i$ and $R'_i$ are calculated with Equation (7) and Equation (8).

1. Terminate the rays that reach the recursive depth $D_{\max}$.

2. Evaluate the intersection status of each ray with the object, categorizing it into three scenarios: non-intersection, intersection from outside the object, and intersection from inside the object.

3. For rays that do not intersect with the object, return the color rendered through the environment field.

4. For each ray that intersects with the object externally, an additional ray is emitted from the intersection point $x_i'$ in both the reflection and refraction directions, and these rays are recursively traced. The radiance of the reflected and refracted rays are then blended using the values of $R_i'$ and $T_i'$ computed from Equation (7) and Equation (8). Finally, the resulting radiance $L_i'$ is mixed with the environmental radiance prior to the intersection point. The IoR of the objects is differentiable during this process.

5. For each ray that intersects with the object from the interior, follow the steps in 4), and the radiance is attenuated based on the accumulated absorption rate from the origin to the intersection point, as formulated in Equation (11).

**Differentiable Mesh Intersection.** To further refine the initial mesh obtained in the previous stage, we calculate the intersection between the rays and the mesh in a differentiable manner. Our mesh $\mathcal{M}$ is formulated as $\mathcal{M} = (V, E, F)$ comprising vertices $V$, edges $E$, and faces $F$. The normal $\mathbf{n}$ of the intersecting point is barycentrically interpolated from the vertex normals, which are computed as the average of the face normals:

$$\mathbf{n}(\mathbf{x}) = \sum_{\mathbf{v}_i \in f(\mathbf{x})} \beta_i \mathbf{n}_{v_i}, \tag{12}$$

$$\mathbf{n}_{v_i} = \frac{1}{3} \sum_{f_j \in f, \mathbf{v}_i \in f_j} \frac{(\mathbf{v}_{f_j,1} - \mathbf{v}_{f_j,0}) \times (\mathbf{v}_{f_j,2} - \mathbf{v}_{f_j,0})}{\| (\mathbf{v}_{f_j,1} - \mathbf{v}_{f_j,0}) \times (\mathbf{v}_{f_j,2} - \mathbf{v}_{f_j,0}) \|_2}, \tag{13}$$

where $f(\mathbf{x})$ is the first triangle that the ray intersects, $\mathbf{v}_{f_j,k}$ is the $k$-th vertex in triangle $f_i$, and $w_i$ is the barycentric coordinate of $\mathbf{x}$ in $f(\mathbf{x})$. The intersection points are also interpolated from vertex positions to provide the gradient for the mesh. The illustration is shown in Figure 3.

**Optimization.** The mesh vertices, inner absorptive rate and IoR is jointly optimized in this stage. In order to avoid the influence of over-absorbed pixels, we supervise the rendered color $\hat{\mathbf{c}}$ using $L_2$ loss, with an additional $\mathbf{c}_i$:

$$\mathcal{L}_{\text{color}} = \frac{1}{|\mathcal{B}|} \sum_{i \in \mathcal{B}} \| (\hat{\mathbf{c}}_i - \mathbf{c}_i) \cdot \mathbf{c}_i \|_2^2, \tag{14}$$

where $\mathcal{B}$ denotes the batch size, and $\mathbf{c}_i$ is the ground truth color. Meanwhile, we observed that refracted background may lead to incorrect gradient for the absorption rate due to different darkness. To mitigate the issue, we employ the following regularization to constrain the tone, i.e. the ratio of the three channels in the color space:

$$\mathcal{L}_{\text{tone}} = \left( 1 - \frac{\hat{\mathbf{c}} \cdot \mathbf{c}}{\|\hat{\mathbf{c}}\| \|\mathbf{c}\|} \right)^2 - \text{var}(\mathbf{c}), \tag{15}$$

where $\text{var}(\mathbf{c})$ denotes the variance of $\mathbf{c}$. And we adopt AdamUniform (Nicolet et al., 2021) to optimize the mesh to reduce noises during training. Additionally, to alleviate the bias caused by incorrect background, we adapt local smoothness for the absorption rate on $N$ randomly sampled points $\{\mathbf{v}_i\}_{i=0}^{N-1}$ with random perturbation $\xi_i$:

$$\mathcal{L}_{\text{mat-smooth}} = \frac{1}{N} \sum_{\mathbf{v}_i} |\mu_t(\mathbf{v}_i) - \mu_t(\mathbf{v}_i + \xi_i)|. \tag{16}$$

The total training loss is defined as the weighted sum of all individual losses in the form of:

$$\mathcal{L}_{\text{total}} = \lambda_1 \mathcal{L}_{\text{color}} + \lambda_2 \mathcal{L}_{\text{tone}} + \lambda_3 \mathcal{L}_{\text{mat-smooth}}, \tag{17}$$

where $\lambda_1$, $\lambda_2$ and $\lambda_3$ are balancing weights for the different terms. In addition, we impose regularization terms to mitigate the abrupt variations in the absorption rate, ensuring geometry smoothness and mask consistency. These regularization are applied periodically during training, with further details provided in Section A.2 of the appendix.

## 4 EXPERIMENT

We evaluate our method on both synthetic and real-world datasets. It is noteworthy that, in contrast to previous methods, our *DiffTrans* can not only reconstruct the geometry of transparent objects but also recover the absorptive material properties. To demonstrate the effectiveness of our

approach and the importance of modeling transparency effects, we compare the reconstructed geometry quality with other methods, such as NeRO (Liu et al., 2023), NU-NeRF (Sun et al., 2024) and NeRRF (Chen et al., 2023). Given that our *DiffTrans* is capable of recovering both the geometry and materials of the transparent objects with complex textures, it enables scene editing, such as relighting. Consequently, we also compare the relighting results with those produced by the baseline methods. The implementation details of our approach are included in Section A.3 of the appendix.

## 4.1 DATASETS

For synthetic data, we use two scenes (*bunny* and *cow*) from NEMTO (Wang et al., 2023) dataset for reconstruction of objects without materials, and four scenes (*monkey*, *horse*, *hand* and *mouse*) from (Lyu et al., 2020; Li et al., 2020) for reconstruction of objects with materials. Due to the absence of released rendered images from NEMTO, we follow its configuration by utilizing infinite-distance environment maps and absorption-less glass materials to render a set of multi-view images. Besides, we made a series of real-world dataset (flower, etc.) by capturing a video around the object and its environment using an iPhone in a continuous one-shot style and extracted the frames using ffmpeg. The camera poses are obtained via COLMAP (Schönberger et al., 2016;

| Method | NeRO | NU-NeRF | NeRRF | Ours(S1) | Ours |
|---|---|---|---|---|---|
| $CD(\times 10^{-4})\downarrow$ | 36.022 | 7.891 | 13.341 | 4.666 | **3.264** |
| $F_1(\times 10^{-1})\uparrow$ | 5.691 | 8.026 | 6.916 | 8.088 | **8.386** |

Table 1: Average quantitative comparison of geometry reconstruction across all synthetic datasets. Ours(S1) refers to the coarse geometry after the initialization.

| Dataset | horse | monkey | bunny | cow | hand | mouse |
|---|---|---|---|---|---|---|
| Ours | 1.540 | 1.577 | 1.471 | 1.472 | 1.484 | 1.462 |
| GT | 1.450 | 1.600 | 1.485 | 1.472 | 1.485 | 1.375 |

Table 2: Comparison of the IoR predicted by our *DiffTrans* with the ground truth across 6 synthetic scenes.

Schönberger & Frahm, 2016). We make a utility tool to semi-automatically annotate the masks with SAM (Kirillov et al., 2023), and classify the images into two parts, used to reconstruct the object and the environment respectively, based on the mask proportion in the image. More details and visualization of our dataset are provided in Section A.4 and Section A.5 the appendix.

## 4.2 RESULTS OF RECONSTRUCTION

To validate the quality of the reconstructed geometry produced by our *DiffTrans*, we conducted experiments in 6 synthetic scenes, with quantitative and qualitative comparisons presented in Table 1 and Figure 4, respectively. We adopt the Chamfer distance (CD) and $F_1$-score ($F_1$) as the evaluation metrics. To the best of our knowledge, existing methods struggle to reconstruct transparent objects with intricate textures, resulting in suboptimal geometry quality. For example, the *horse* reconstructed by NeRRF exhibits noticeable surface roughness, while the

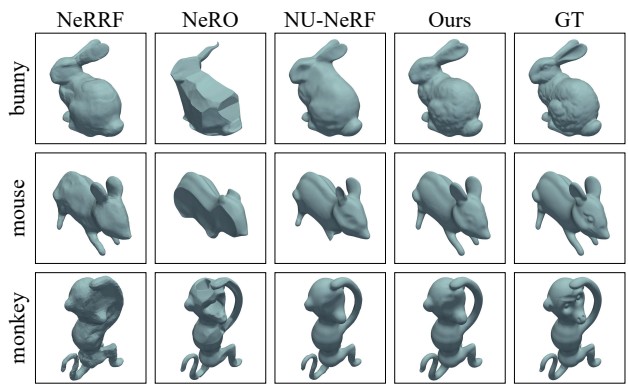

Figure 4: Qualitative comparisons of the reconstructed geometry on *bunny*, *horse* an *monkey* scenes.

void between the hand and head of the *monkey* is erroneously filled. Furthermore, the *bunny* and *monkey* reconstructed by NeRO display structural inaccuracies. Critically, our *DiffTrans* achieves superior performance, outperforming all baselines in average CD and $F_1$-score across all the scenes. In the second stage, while optimizing the materials of the transparent objects, Our *DiffTrans* further refine the geometry obtained from the first stage. As evidenced by the last two rows in Table 1, the refined geometry demonstrates measurable quality improvements, substantiating the effectiveness of our proposed differentiable recursive mesh ray tracer.

In contrast to most approaches that rely on predefined IoR, our *DiffTrans* directly optimizes the IoR in the second stage. Table 2 shows the IoR predicted by our *DiffTrans* across 6 synthetic scenes,

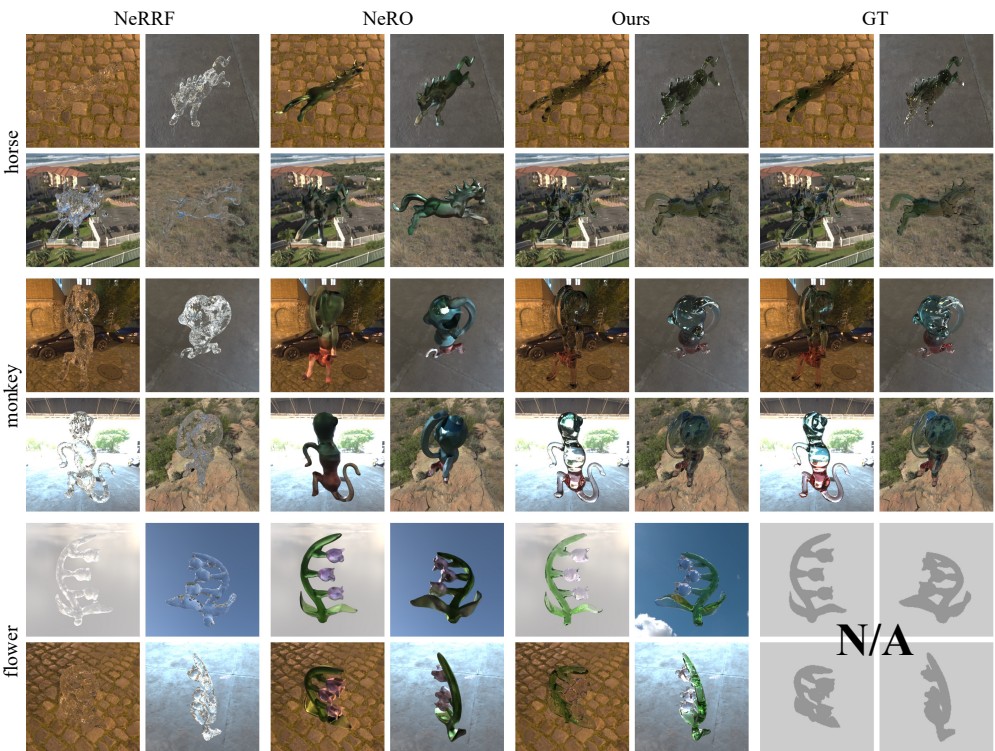

Figure 5: Qualitative comparison of relighting results on the *horse*, *monkey*, and *flower* scenes.

| Dataset | horse | | | monkey | | | flower | | | hand | | | mouse | | |
|---------|-------|-------|--------|--------|-------|--------|--------|-------|--------|-------|-------|--------|-------|-------|--------|
| | PSNR↑ | SSIM↑ | LPIPS↓ | PSNR↑ | SSIM↑ | LPIPS↓ | PSNR↑ | SSIM↑ | LPIPS↓ | PSNR↑ | SSIM↑ | LPIPS↓ | PSNR↑ | SSIM↑ | LPIPS↓ |
| w/o $\mathcal{L}_{tone}$ | 27.00 | 0.9208 | 0.0514 | 24.15 | **0.8942** | 0.0663 | 24.06 | 0.7883 | 0.2731 | 22.37 | 0.8719 | 0.1481 | 24.15 | 0.8980 | 0.1225 |
| Full | **27.03** | **0.9278** | **0.0457** | **24.62** | 0.8935 | **0.0631** | **24.14** | **0.7895** | **0.2717** | **23.39** | **0.8736** | **0.1420** | **24.16** | **0.8986** | **0.1219** |

Table 3: Quantitative results of novel view synthesis with/without the tone regularization.

where it can be observed that the discrepancies between the predicted results and the ground truth IoR are minimal. The complete qualitative results of geometry and novel view synthesis are provided in Section A.5 of the appendix.

## 4.3 RESULTS OF RELIGHTING

In addition to the 5 scenes with complex textures (*monkey*, *horse*, *flower*, *hand* and *mouse*), we also compare the results of relighting on two scenes from NEMTO (*Bunny* and *cow*). To ensure a fair comparison, we set a fixed index of refractive (IoR) for the two NEMTO scenes to align with the settings used in NeRRF (Chen et al., 2023). For each scene, we employ 5 distinct environment maps for illumination and subsequently render images by randomly selecting 10 camera viewpoints that collectively cover the upper hemisphere centered on the object, ensuring comprehensive validation.

We compare our *DiffTrans* with NeRRF (Chen et al., 2023), current state-of-the-art method for transparent object reconstruction, and NeRO (Liu et al., 2023), the recent method for inverse rendering. Since NU-NeRF (Sun et al., 2024) employs the implicit representation, we are unable to directly obtain its relighting results. Table 4 shows the quantitative results of relighting using the met-

| Method | PSNR↑ | SSIM↑ | LPIPS↓ |
|--------|-------|-------|--------|
| NeRRF | 19.25 | 0.8118 | 0.0812 |
| NeRO | 19.64 | **0.8500** | 0.0856 |
| Ours | **23.17** | 0.8380 | **0.0678** |

Table 4: Quantitative results of relighting averaged across all the synthetic scenes.

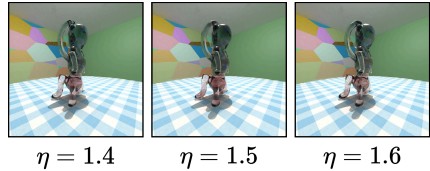

$\eta = 1.4$     $\eta = 1.5$     $\eta = 1.6$

Figure 6: Reconstruction results with different IoRs on the *monkey* scene.

rics including PSNR, SSIM and LPIPS averaged across all scenes. Our *DiffTrans* outperforms other methods on most metrics, demonstrating the superiority of our approach. We also present the qualitative comparisons in Figure 5. Due to NeRRF's inability to reconstruct the materials of transparent objects and its relatively low geometry quality, the resulting relighting outcomes are subpar. NeRO focuses solely on reflection and fails to simulate the refraction process, leading to unrealistic relighting results. In contrast, our *DiffTrans* achieves high-quality reconstruction of both geometry and materials, enabling it to achieve plausible relighting.

### 4.4 ABLATION STUDIES

We conduct ablation experiments to explore the effectiveness of the tone regularization introduced in Equation (15). As shown in Table 3, while the regularization leads to a slight decrease in the SSIM metric for novel view synthesis in the *monkey* scene, all other metrics have shown improvement. Figure 6 shows that our method is robust under different IoR settings.

Please refer to Section A.6 of the appendix for ablation studies for other components of our method, and the qualitative results of the ablation study of the tone regularization.

## 5 CONCLUSIONS

In this work, we have presented a differentiable rendering framework, *DiffTrans*, for reconstructing transparent objects with diverse geometry and complex texture. The proposed *DiffTrans* first reconstructs the initial geometry and the environment separately. Then it performs joint optimization over the geometry, index of refraction and absorption rate of the transparent objects in a unified and end-to-end manner, using the specifically designed recursive differentiable ray tracer. In particular, the ray tracer can be implemented efficiently in CUDA to reduce computational cost. Extensive experiments have demonstrate the effectiveness of our method, especially towards transparent objects with intricate topology and texture.

**Limitations.** We have made three assumptions during the problem formulation, as described in Section 3.1. Despite their correctness in most typical scenes, these assumptions would inevitably introduce modeling inaccuracies in some complicated scenes. We will investigate further improvement to relax the assumptions, thereby improving the robustness of our method in future works.

### ACKNOWLEDGEMENTS

This work was supported by the National Natural Science Foundation of China (Grant No. 62372133, 62125201 and U24B20174).

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

# A APPENDIX

## A.1 DETAILS ON GEOMETRY INITIALIZATION

Geometry initialization constitutes a critical first step in our proposed method, aiming to derive an initial geometry estimation solely from object masks and camera parameters to enhance training stability. This approach is designed to achieve the following key objectives:

- **Mask Consistency**. The initial geometry must align closely with the input mask, regardless of its shape. Complex topologies within the mask can sometimes fail to provide adequate visibility gradients, particularly for rasterization-based methods. Incorporating an MLP to model spatially-varying data can enhance smoothness and improve visibility gradients (Mehta et al., 2022). However, this approach significantly reduces training efficiency and limits the model's generalizability for intricate structures. Fully implicit methods that rely on MLP optimization may encounter discretization errors and fractured topologies, while explicit methods often struggle to handle topology changes during training without imposing additional constraints (Mehta et al., 2022).

- **Interpolation Capability**. The initial geometry must be sufficiently smooth, particularly in regions within the image mask that lack detailed information. For occluded areas, the image mask does not provide adequate information, necessitating reasonable interpolation to fill these gaps. Methods such as space carving (Mehta et al., 2023) can effectively generate meshes that align with the mask, but they are unable to perform interpolation in occluded regions.

- **Speed**. Due to the limited information provided by masks, prolonged initialization does not necessarily yield better results. Implicit methods which use MLP for prediction typically exhibit slower training speed and often lead to overly smooth surfaces, while explicit methods such as space carving tend to have poorer generalization capabilities. Therefore, it is crucial to achieve a balance between implicit field optimization and explicit algorithms to ensure both speed and adaptability.

Based on the considerations outlined above, we adopt a simple yet efficient approach for initialization. First, we utilize FlexiCubes as the iso-surface representation, which is optimized by rasterizing the extracted differentiable mesh and minimizing the loss between the rendered results and the ground truth mask. To mitigate potential training instabilities and noise arising from the ill-posed nature of the mask, we impose constraints on smoothness, mesh quality, and SDF during training. This process is both fast and efficient, enabling the geometry to adapt to various topologies while maintaining sufficient smoothness to provide a robust starting point for optimization. However, this dense representation lacks precision and contains significant redundant information. To address these limitations, we extract the mesh from the trained FlexiCubes and remesh it to achieve a more detailed and uniform structure. The refined mesh is further optimized using the mask to improve mask consistency. This enhanced mesh serves as the foundation for subsequent optimization steps.

## A.1.1 INITIALIZATION-PHASE MESH REPRESENTATION

At the initialization stage, our mesh is defined within a Signed Distance Field (SDF) $f(\mathbf{x})$. Specifically, the mesh is obtained by extracting an iso-surface $\mathcal{S} = \{\mathbf{x}|f(\mathbf{x}) = 0\}$ within it. We adopt FlexiCubes (Shen et al., 2023) for geometry initialization at the starting stage, where the SDF values are stored in the uniform cubic grid vertices inside. A mesh can be extracted by Dual Marching Cubes (Shen et al., 2023), where a primal mesh is firstly extracted by Marching Cubes, and the final mesh is generated such that all its vertices lies on unique faces of the primal mesh. We denote the final mesh as:

$$M = (V, E, F), \tag{18}$$

where $V = \{v|v \in \mathbb{R}^3\}$ represents all vertices in the mesh, $E = \{\{v_i, v_j\}|v_i, v_j \in V\}$ is the unique edge set, and $F = \{\{v_i, v_j, v_k\}|v_i, v_j, v_k \in V\}$ is the set of unique faces. As vanilla FlexiCubes are limited in a bounded region, we provide a AABB bounding box $(\mathbf{x}_{\min}, \mathbf{x}_{\max})$ of the estimated object area as the input of the initialization, which is easily obtainable by the MVS progress of creating the dataset or AABB boxes on several images manually annotated by users. Vertices $\{v_i\}$ of the mesh

are transformed as:

$$v_i' = v_i \circ (\mathbf{x}_{\max} - \mathbf{x}_{\min}) + \mathbf{x}_{\min} \tag{19}$$

from the original $[-0.5, 0.5]^3$ region.

### A.1.2 OPTIMIZATION

At each training iteration, we obtain the object's mesh from its grid representation, then we apply nvdiffrast (Laine et al., 2020) for differentiable rasterization at training views. Concretely, the mask is normally rasterized, and is antialiased with the triangle edges to provide visibility gradient w.r.t the vertex positions. After obtaining the antialiased mask $\hat{\mathcal{M}}$, we supervise it with the ground truth mask $\mathcal{M}$:

$$\mathcal{L}_{\text{geo-init}} = \frac{1}{N} \sum_{i \in \mathcal{B}} \|\hat{\mathcal{M}}_i - \mathcal{M}_i\|_1, \tag{20}$$

where $\mathcal{B}$ denotes the batch size and $N$ represents the number of pixels.

**SDF Dilation.** Our geometry is determined by the visibility gradient, which is derived from the differences in masks at the edges of the rendered images. The anti-aliased mask values of the edge pixels are calculated based on the intersection points between the projected triangle edges and the lines connecting the centers of the pixels. These differences in mask values directly contribute to the gradients at the edge positions, thereby influencing the locations of the vertices. Since the gradient is only applied to the edges, achieving proper topology fitting relies on dispersing the edges across a wide area. To achieve this, we adopt the approach proposed in (Munkberg et al., 2022) and initialize the vertex SDF with a uniform distribution $U[-0.1, 0.9]$, ensuring that our initial geometry is evenly scattered within our area of interest. However, relying solely on the mask loss for geometry recovery leads to poor internal structure. The resulting mesh may have the correct mask but is distorted and noisy internally. In order to address this issue, an alternative approach involves starting with a cleaner initial mesh, such as a sphere (Chen et al., 2023), instead of random initialization. While this approach fixes the broken topology within the mask, it also imposes limitations on the range of topologies that can be accommodated.

To fill the cracks of the mesh, we dilate its surface with the following loss:

$$\mathcal{L}_{\text{dilation}} = \frac{1}{|V|} \sum_{v \in V} f(v). \tag{21}$$

The regularization tends to decrease the SDF values, expanding the iso-surface and filling the gaps. The utilization of this operation is based on two key observations. Firstly, the areas within the mask do not receive any gradient from the mask loss $\mathcal{L}_{\text{geo-init}}$ and therefore remain stable after being filled. Secondly, the cracks within the mask are smaller in scale compared to the geometry of the ground truth shape. This implies that the dilation operation will not obscure genuine geometry details. It is important to note that the scalar values stored in the FlexiCubes grid do not represent true SDF, which exhibit a consistent absolute value of spatial gradient. Instead, they serve as indicator functions that are flat in empty and full spaces, respectively. Consequently, Equation (21) not only dilates the mesh within the mask, but also generates new geometry throughout the entire space. We consider this to be a beneficial feature that aids in capturing missed geometry.

**Mesh Regularization.** While the SDF dilation technique successfully recovers the correct object topology, the resulting surface remains rough and exhibits high-frequency noise. In order to mitigate these issues, we employ a rendering approach to generate the mesh's normal map $\mathbf{n}$ and depth map $d$. We then introduce a smoothness regularization term that operates on the derivatives in screen space:

$$\mathcal{L}_{\text{smooth}} = \frac{1}{N} \sum_i \left( \|\nabla d_i\|_1 + \|\nabla \mathbf{n}_i\|_1 + \|\nabla^2 \mathbf{n}_i\|_1 \right), \tag{22}$$

where $\nabla$ denotes screen space gradient along the direction, and $N$ represents the pixel count over the whole batch. The derivatives in Equation (22) are computed using screen space finite differences. This screen space regularization can be applied to random views. We have noticed that for objects with intricate details, such as holes, that appear small within the view, the mask loss alone may not entirely eliminate all the floating artifacts. To address this issue, we adopt the approach proposed in (Munkberg et al., 2022) and employ a binary cross entropy function with logistic activation

`BCELogit`$(x, y) = -[y \log(\sigma(x)) + (1 - y) \log(1 - \sigma(x))]$ to eliminate sparse floating artifacts among neighboring vertices with different SDF signs:

$$\mathcal{L}_{\text{BCE}} = \frac{1}{|E_f|} \sum_{\substack{(v_i, v_j) \in E_f \\ v_i \neq v_j}} \text{BCELogit}(f(v_i), f(v_j)), \tag{23}$$

in which

$$E_f = \{(v_i, v_j) | v_i, v_j \in E \wedge \text{sign}(f(v_i)) \neq \text{sign}(f(v_j))\} \tag{24}$$

is the set of unique and directional edges crossing the extracted mesh face. Due to the dilation process, our recovered mesh may be swelling at unseen faces, so we restrict the total surface area by:

$$\mathcal{L}_{\text{area}} = \frac{1}{|F'|} \sum_{f \in F'} \text{area}(f). \tag{25}$$

Our mask loss penalizes out-of-mask triangles, which may result in excessive gradients at grazing angles and causes pit on the surface. To ensure a more consistent mesh structure and avoid the occurrence of pits on the surface, we introduce a series of regularization techniques to mitigate the excessive gradients that may arise at grazing angles in our mask loss. The first regularization step involves penalizing the average distance between the mesh vertices and their respective primal faces:

$$\mathcal{L}_{\text{avgdis}} = \frac{1}{|V'|} \sum_{v \in V'} \text{MAD}\left(\{\|v - u_e\|_2 | u_e \in \mathcal{N}_v\}\right), \tag{26}$$

where $\mathcal{N}_v$ is the primal face of mesh vertex $v$ and MAD means mean absolute deviation of a set of values to their mean. This will add to its flexibility by leaving places in all deforming directions. Besides that, we also employ mesh developability loss $\mathcal{L}_{\text{dev}}$ introduced in (Stein et al., 2018) to encourage flatness and keep hinges and L1 loss on grid weights $\mathcal{L}_{\text{L1}}$ to avoid local minimum. Performing dilation and smoothing too early may cause the initial mesh wrongly filled because redundant floaters are not cleared by the mask, so we mainly use the mask loss at the early stage. We also adapt a switching scheme between the dilation and smoothing for training efficiency. The overall loss for the geometry initialization can be written as:

$$\begin{aligned} \mathcal{L}_{\text{geo}} = &\lambda_1 \mathcal{L}_{\text{geo-init}} + \lambda_2 \mathcal{L}_{\text{dilation}} + \lambda_3 \mathcal{L}_{\text{smooth}} + \lambda_4 \mathcal{L}_{\text{BCE}} \\ &+ \lambda_5 \mathcal{L}_{\text{area}} + \lambda_6 \mathcal{L}_{\text{avgdis}} + \lambda_7 \mathcal{L}_{\text{dev}} + \lambda_8 \mathcal{L}_{\text{L1}}. \end{aligned} \tag{27}$$

The regularization requires only a little hyper-parameter tuning across datasets and is efficient to train. The scheme details are shown in Section A.3.2.

**Post-processing.** After receiving the mesh from FlexiCubes, to compensate for the over-smooth effect caused by the screen space smoothness regularization, we additionally use Botsch-Kobbelt method (Botsch & Kobbelt, 2004) to remesh and optimize it using AdamUniform optimizer (Nicolet et al., 2021) with edge normal regularization:

$$\mathcal{L}_{\text{edge}} = \frac{1}{|E'|} \sum_{\{v_i, v_j\} \in E'} \left(1 - \mathbf{n}_{v_i} \cdot \mathbf{n}_{v_j}\right)^2. \tag{28}$$

## A.2 Additional Details on the Ray Tracer

The recursive ray tracer is leveraged to optimize the geometry, index of refraction and absorptive rate of the objects. In addition to the methods described in the main paper, we incorporated several regularization strategies during training to further enhance the model's generalization ability and robustness.

### A.2.1 Optimization

**Volume Regularization.** As discussed in the main paper, our method models the material of transparent objects explicitly by directly render under the Lambert-Beer's law, which is a highly ill-posed process. To ensure the model's robustness and avoid being trapped in local minima, we introduce a $L_2$ penalization term on the volume density $\mu_t$ of the object. The regularization term is defined as:

$$\mathcal{L}_{\text{vol}} = \sum_{i=1}^{N} \|\mu_t(\mathbf{x}_i)\|_2^2, \tag{29}$$

where $\mathbf{x}_i$ is the $i$-th point sampled randomly from the volumetric material field.

**Mesh Regularization.** Due to the unstable nature of the rendering of transparent objects, we apply mask regularization to the mesh to preserve mesh consistency. The regularization term is defined as:

$$\mathcal{L}_{\text{mask}} = \sum_{i=1}^{N} \left\| \hat{\mathcal{M}}_i - \mathcal{M}_i \right\|_1, \tag{30}$$

where $\mathcal{M}_i$ is the $i$-th rendered mask in $N$-sized batch, and $\hat{\mathcal{M}}_i$ is the corresponding ground truth mask.

**Smoothness Regularization.** We observed that the mesh exhibits noises during the training process, which may lead to artifacts in the rendered images. To alleviate this issue, we introduce a smoothness regularization term on the mesh vertices by smoothing the vertex normals. The regularization term is defined as:

$$\mathcal{L}_{\text{edge}} = \frac{1}{|E'|} \sum_{\{v_i, v_j\} \in E'} \left( 1 - \mathbf{n}_{v_i} \cdot \mathbf{n}_{v_j} \right)^2, \tag{31}$$

where $E'$ is the set of edges in the mesh, and $\mathbf{n}_{v_i}$ is the normal of vertex $v_i$.

**Periodical Regularization.** To ensure the model's robustness and avoid overfitting, we introduce a periodical regularization term on the geometry and material parameters. We regularize the mesh with $\mathcal{L}_{\text{mask}}$, $\mathcal{L}_{\text{edge}}$ and Laplacian regularization $\mathcal{L}_{\text{lap}}$ to improve he quality of the mesh every $n$ iterations. The Laplacian regularization is detailed in (Nicolet et al., 2021) to control the uniformity of the mesh.

**Overall Loss.** The overall loss function for the ray tracer is defined in the form of summation of the loss $\mathcal{L}_{\text{total}}$ introduced in the main paper and the regularizations:

$$\mathcal{L}_{\text{rt}} = \mathcal{L}_{\text{total}} + \lambda_4 \mathcal{L}_{\text{vol}} + \lambda_5 \mathcal{L}_{\text{mask}} + \lambda_6 \mathcal{L}_{\text{edge}}. \tag{32}$$

## A.3 Implementation Details

### A.3.1 Environment Details

We conducted our experiments on a machine with an Intel Core i7-10700K CPU and 64GB of RAM, with a single RTX 3090 GPU. The software environment includes Python 3.11, PyTorch 2.4, CUDA 12.6 and Ubuntu 24.04. We compiled our ray tracer with g++ 13 and nvcc 12.6.

### A.3.2 Training Scheme for Geometry Initialization

We employ the Adam optimizer with parameters $\beta_1 = 0.9$, $\beta_2 = 0.999$, and a learning rate $\gamma$ defined as $0.01 \times 10^{-0.0002i}$, where $i$ represents the iteration count. Given the complexity of directly optimizing the explicit geometry, we propose a carefully structured training scheme during the initialization phase, spanning 1000 iterations. This process is broadly divided into three distinct stages:

- **Mask Regression.** At the outset, the FlexiCubes representation begins in a random state. To ensure sufficient visibility gradients from the random edges while minimizing redundant geometry, we apply minimal regularization during this stage. By the end of this phase, the geometry concentrates within the mask-indicated region, albeit with significant noise.
- **Surface Reconstruction.** This stage focuses on refining the geometry into a complete surface. We achieve this by primarily employing dilation loss and smoothness loss, which help form a continuous mesh while maintaining geometric smoothness.
- **Quality Improvement.** Finally, the quality of the geometry is enhanced further by incorporating additional losses, such as developability loss, to refine and improve the surface structure.

Specifically, let $i$ denote the current iteration. Throughout all stages, we set $\lambda_5 = 2$, $\lambda_6 = 0.5$, $\lambda_8 = 1$, and $\lambda_4 = 0.2\,(1 - 0.00076i)$. During the initial 100 iterations, we define $\lambda_1 = c_1$,

| Dataset | horse | monkey | flowers | bunny | cow |
|---------|-------|--------|---------|-------|-----|
| $c_1$ | 7 | 5 | 3 | 5 | 5 |
| $c_2$ | 0.5 | 0.1 | 0.3 | 0.5 | 0.1 |

Table 5: Hyperparameters for geometry initialization across datasets.

| Dataset | horse | monkey | flowers |
|---------|-------|--------|---------|
| $\gamma_3$ | 0.0001 | 0.01 | 0.01 |
| $k$ | 300 | 500 | 1000 |

Table 6: Hyperparameters for freeze-geometry stage.

$\lambda_2 = 0.2c_2$, and $\lambda_3 = \lambda_7 = 0$, where $c_1$ and $c_2$ are hyperparameters. After the first 100 iterations, we employ an alternating strategy by adjusting the settings as follows:

$$\lambda_2 = \begin{cases} \max\left\{ \frac{0.7 - i/1000}{0.6}, -0.05 \right\}, & \text{if } i \bmod 50 \text{ is even,} \\ 0, & \text{otherwise,} \end{cases} \tag{33}$$

$$\lambda_3 = \begin{cases} 0.5, & \text{if } i \bmod 50 \text{ is odd,} \\ 0, & \text{otherwise,} \end{cases} \tag{34}$$

$$\lambda_7 = \begin{cases} \max\left\{ i/100 - 7, 0 \right\}, & \text{if } i \bmod 50 \text{ is odd,} \\ 0, & \text{otherwise.} \end{cases} \tag{35}$$

We adjust $c_1$ and $c_2$ across datasets to align with the scale of the geometry, as detailed in Table 5. After completing 1000 iterations, we extract the mesh and refine it using the Botsch-Kobbelt algorithm. Subsequently, we perform an additional 1000 iterations to optimize the mesh. During this phase, we leverage object masks by minimizing the $L_1$ loss between the rendered mask and the ground truth mask, in conjunction with the smoothness loss $\mathcal{L}_{\text{edge}}$ on vertex normals.

### A.3.3 TRAINING SCHEME FOR THE RAY TRACER

To address the potential bias introduced by initializing the material field as fully transparent, given its deviation from the ground truth color of the object, we freeze the object's geometry for the first $k$ iterations, where $k$ is a user-defined parameter. During this stage, we employ the Adam optimizer with $\beta_1 = 0.9$, $\beta_2 = 0.999$, a weight decay penalty of $\lambda = 10^{-6}$, and learning rates of $\gamma = 3 \times 10^{-3}$ for the material network and $\gamma = 10^{-4}$ for the IOR. The loss weights are set as $\lambda_1 = 1$, $\lambda_2 = 0.001$, and $\lambda_4 = 0.0005$. For the NEMTO datasets, this initialization stage is omitted, as these datasets lack absorption properties. Additional hyperparameter details are provided in Table 6.

Following the freeze-geometry stage, we optimize the geometry and material fields jointly. The same Adam optimizer configuration is used, with the exception that the learning rate for the IoR is updated to $\gamma = 10^{-3}$. For mesh optimization, we employ the AdamUniform optimizer (Nicolet et al., 2021) with a learning rate of $\gamma = 0.001$. To ensure the mesh adheres to the object mask and maintains high geometry quality, we perform 200 iterations of optimization, applying mask constraints and geometry quality regularization as described in Section A.3.2, with these constraints enforced every 100 iterations.

### A.3.4 COMPUTATIONAL AND TRAINING EFFICIENCY

All experiments were conducted on NVIDIA RTX 3090 GPUs with 24 GB of VRAM. The training time for our scenes fell within 1–2 hours, depending on the geometric complexity of the object. Higher geometric complexity requires more ray bounces for a fixed batch size, which in turn increases the per-iteration training time. In our implementation, rays with more than four bounces are discarded during training. For inference, this threshold is increased to eight to achieve better visual quality. The memory consumption is approximately 20 GB during training with a batch size of 5,000 rays, and it never exceeded the available VRAM even in our most complex scene.

### A.3.5 Dynamic Range of Input Images

We use HDR environments for synthetic scenes. For real-world captures with LDR inputs, we apply gamma correction with $\gamma = 2.2$ to mitigate the domain gap between LDR input images and HDR environment maps.

## A.4 Details on Datasets

As detailed in the main paper, our dataset comprises five scenes: *monkey*, *horse*, *flower*, *bunny*, and *cow*. The corresponding number of images, material types, and whether the dataset is synthetic are summarized in Table 7. Objects labeled as "N/A" under material do not have specific material attributes assigned. We provide visualizations of the dataset images in Figure 7. All synthetic

| Dataset | Images | | Material | Real-World |
|---|---|---|---|---|
| | Object | Env. | | |
| horse | 100 | 100 | absorption | N |
| monkey | 100 | 50 | absorption | N |
| flower | 104 | 56 | absorption | Y |
| flower(2) | 70 | 12 | absorption | Y |
| flower(3) | 154 | 69 | absorption | Y |
| flower(4) | 119 | 27 | absorption | Y |
| bunny | 100 | - | N/A | N |
| cow | 100 | - | N/A | N |

Table 7: Overview of the datasets used in our experiments.

datasets are rendered using camera views sampled from a uniform distribution, either on the upper hemisphere for custom datasets or the full sphere for NEMTO datasets, with the object at the center. For the real-world dataset (*flower*), images were captured using an iPhone 11 by recording a video around the object and its environment. Individual frames were extracted from the video, and camera poses were estimated using COLMAP (Schönberger & Frahm, 2016). The point cloud generated during the pose estimation process was used to determine the axis-aligned bounding box (AABB) of the transparent object. Masks for the real-world dataset were annotated using a combination of SAM and manual refinements, requiring approximately 30 minutes for the *flower* dataset.

## A.5 More Visualization Results

We present the qualitative comparison of the novel view synthesis results of our method with the ground truth in Figure 9. All images were rendered using our ray tracing renderer, with additional results available in the `nvs` directory. As shown in Figure 9, in the *horse* scene, our method successfully reconstructed the green absorption material and the mesh of the transparent object, demonstrating high fidelity in object recovery. In the *monkey* scene, our method accurately captured the color transition between the upper body and lower part. Additionally, our method achieves high-fidelity reconstruction in the real-world captured *flower* scene.

Besides the NVS results, we also provide visualization for reconstructed geometry in the `geo_vis` directory. Especially, results on *horse*, *monkey* and *bunny* scenes are shown in Figure 10.

We show in Figure 8 an extra real-world test scene including the input images, recovered normal maps and relighting results.

The rendering result between the meshes from the stage 1 and the stage 2 are compared in Figure 16, which demonstrates the effectiveness of our stage-2 refinement.

To show the effectiveness of our stage-2 reconstruction, we provide a rendered heatmap showing the difference in Hausdorff Distance between the ground truth before and after optimization in Figure 11 and the difference between stage-1 and stage-2 meshes in Figure 12.

The figure of recovered scenes are shown in Figure 13.

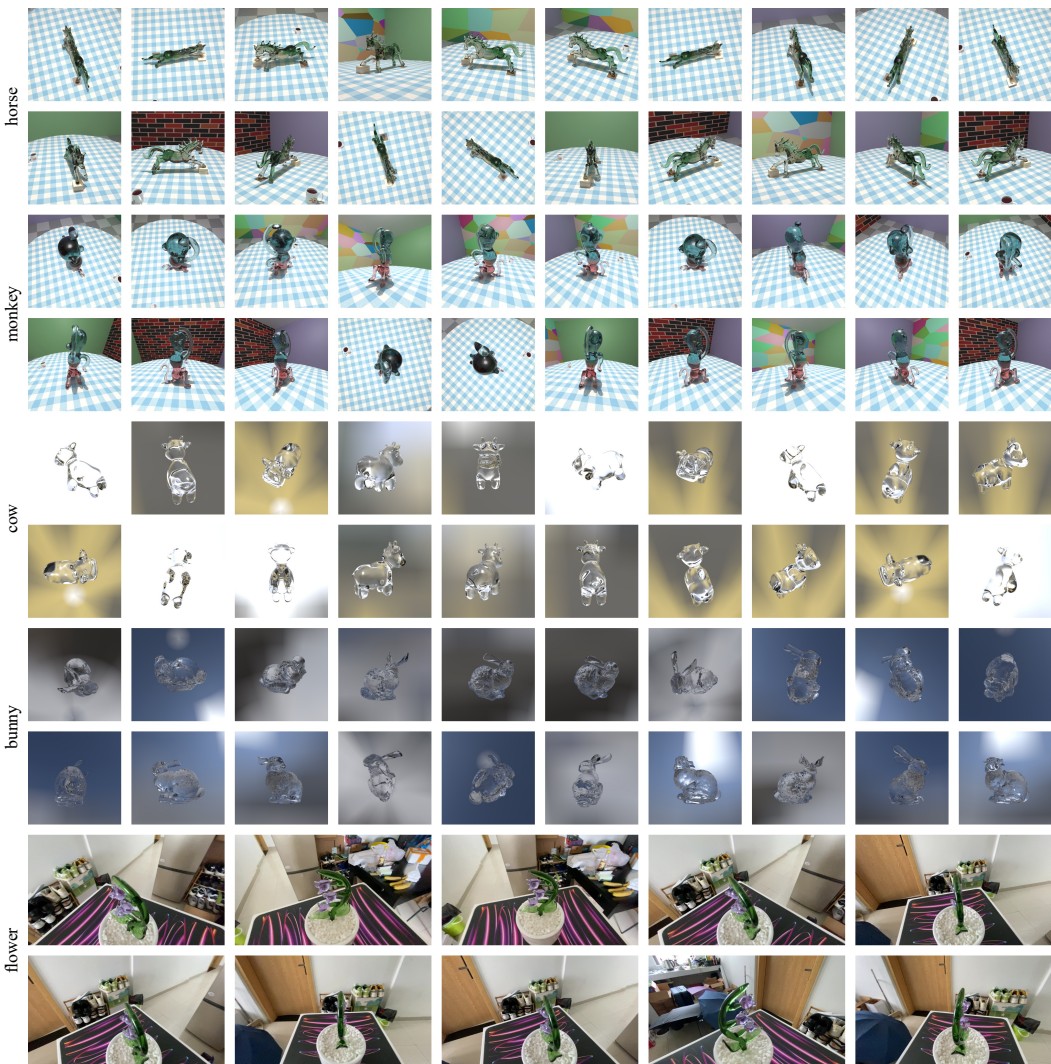

Figure 7: Parts of the images from the dataset used in our experiments.

A ablational novew synthesis result of different IORs on the *monkey* and *horse* scenes are shown in Figure 14.

To better visualize the reconstruction result of the geometry, we provide the normal maps of the reconstructed meshes in Figure 15.

## A.6 MORE ABLATION STUDIES

### A.6.1 ABLATION ON GEOMETRY INITIALIZATION

To evaluate the reliability and effectiveness of the proposed method, we conducted ablation studies on the SDF dilation loss and the smoothness loss introduced in Section A.1 on four meshes: *block*, *frame*, *horse* and *monkey*. All those meshes are rendered with 100 camera views drawn from a uniform distribution on the upper semi sphere centered at the object. As shown in Table 10, the joint usage of our dilation, smoothness and other operations including the original relularizations used in Flexicube has facilitated the reconstruction of high-quality meshes solely from mask supervision. The absence of floater-removing regularization leads to unsuccessful recovery on geometry structures containing voids, i.e., frames and blocks. Qualitative results are shown in Figure 17.

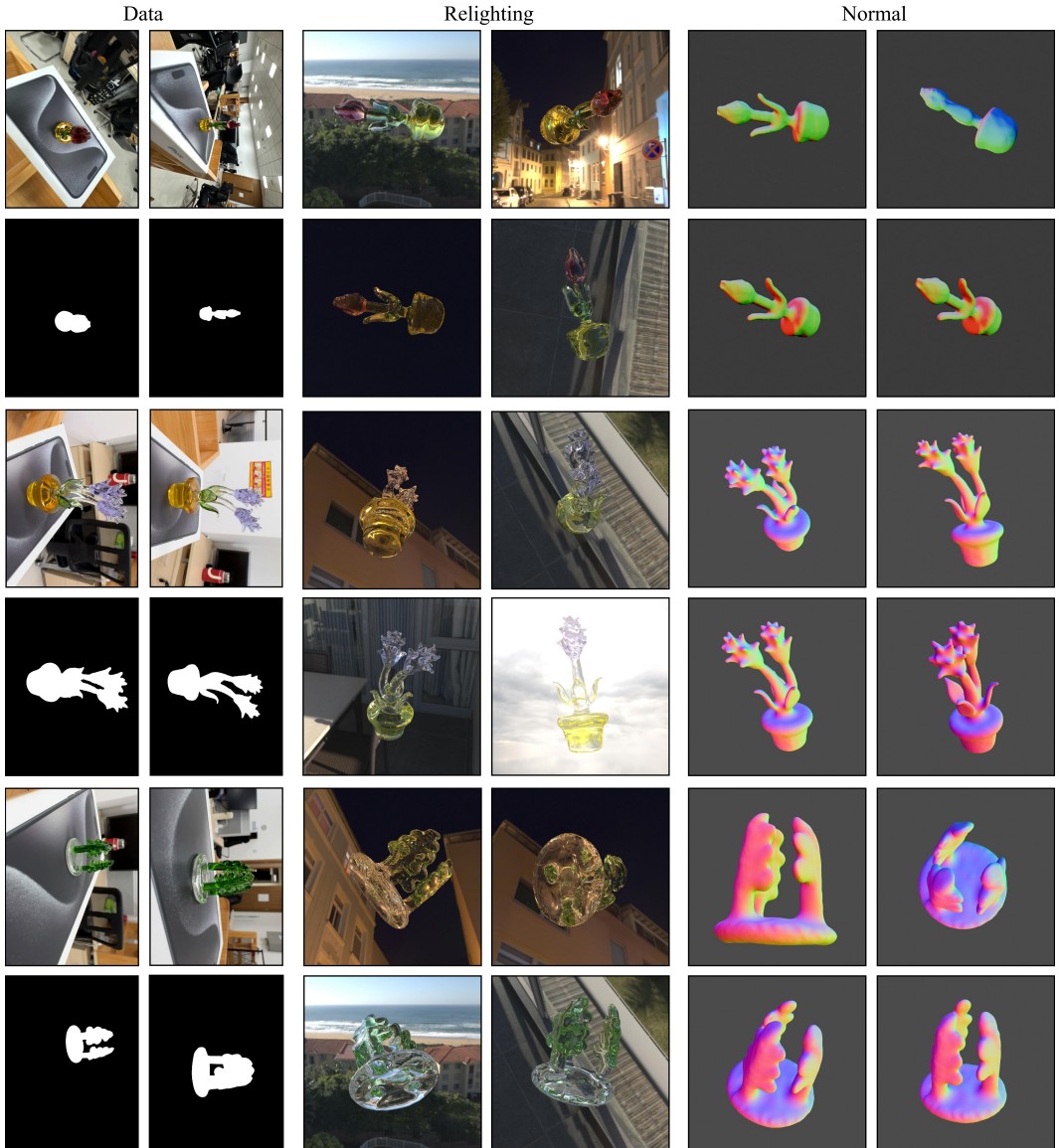

Figure 8: Visualization of an extra real world test scene including the input images, recovered normal maps and relighting results.

### A.6.2 Ablation on Input Mask Noises

To investigate the impact of mask error, we ablations by adding different levels of noise to the object masks. Specifically, we randomly dilate of erode the masks on their edges by 3 pixels or crop off the masks by 1-3 patches with the width of 10 or 20 pixels. The quantitative results are shown in Table 8. The results in Table 8 show that the performance is largely unaffected, with metrics varying only within a small range. This demonstrates the robustness of our method with respect to mask accuracy.

### A.6.3 Ablation on Tone Regularization

Besides, we conduct ablations to evaluate the effectiveness of tone regularization in Equation (15), and the qualitative results are shown in Figure 18.

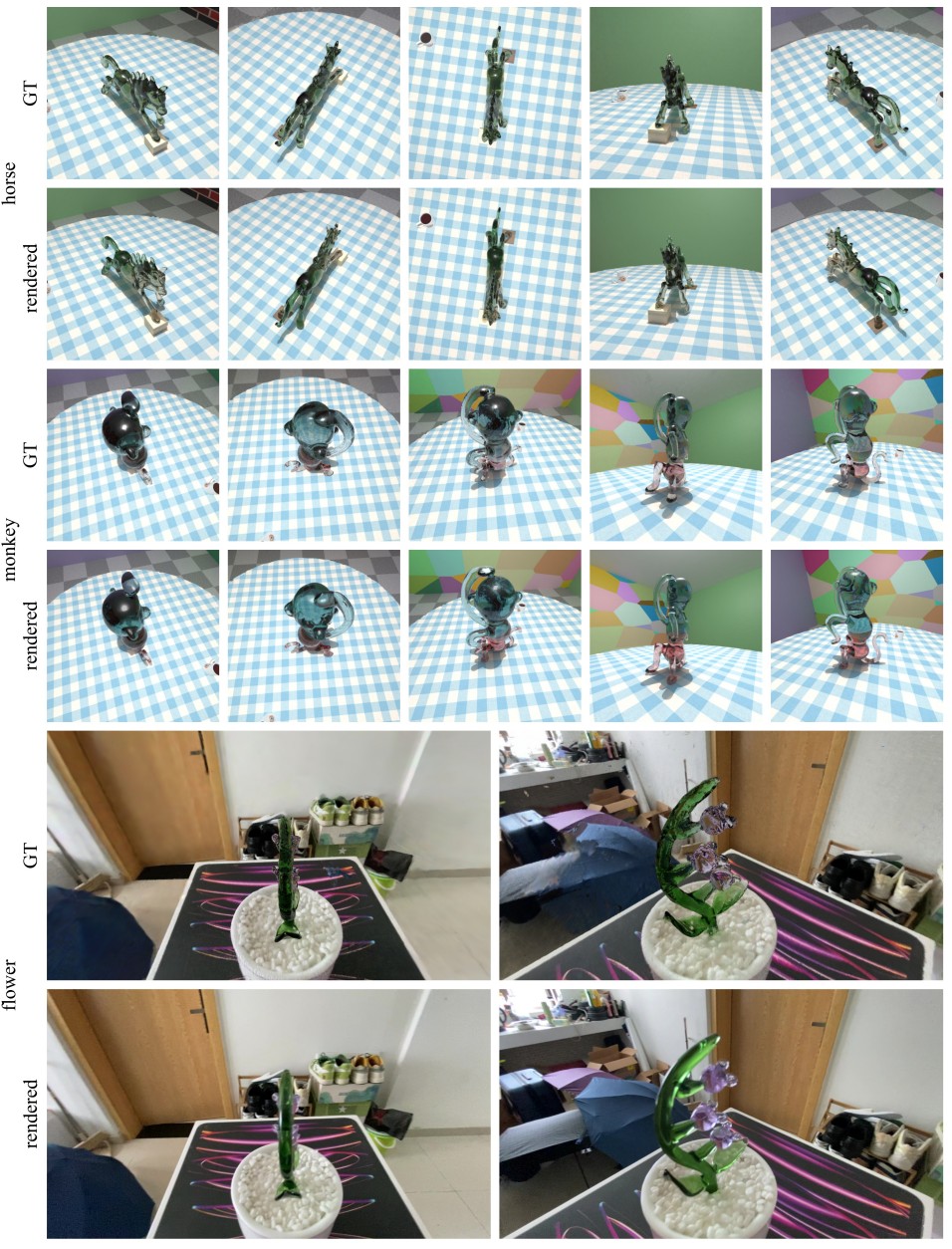

Figure 9: Novel view synthesis results on the *horse*, *monkey* and *flower* scenes.

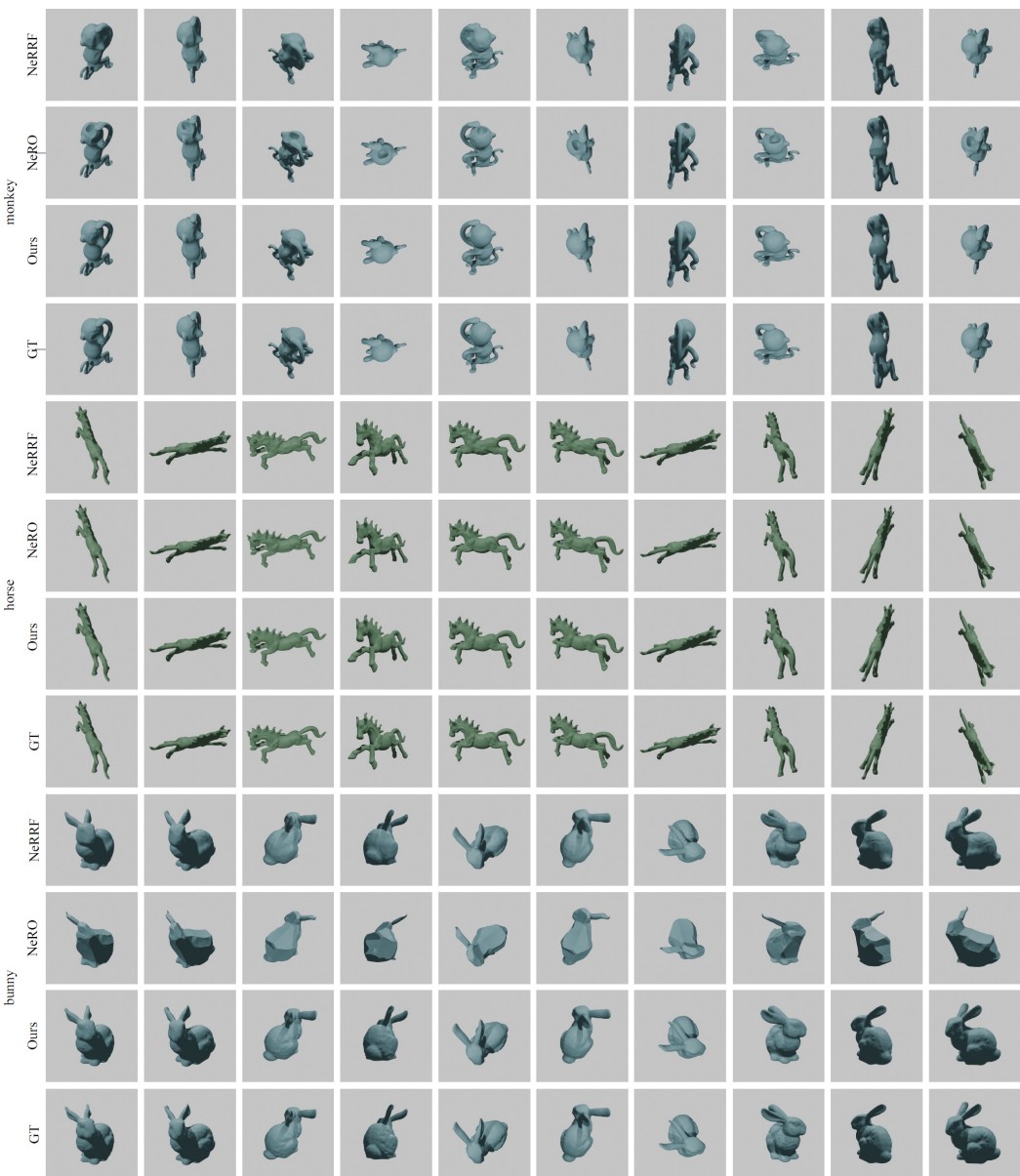

Figure 10: Reconstructed geometries on the *horse*, *monkey* and *bunny* scenes.

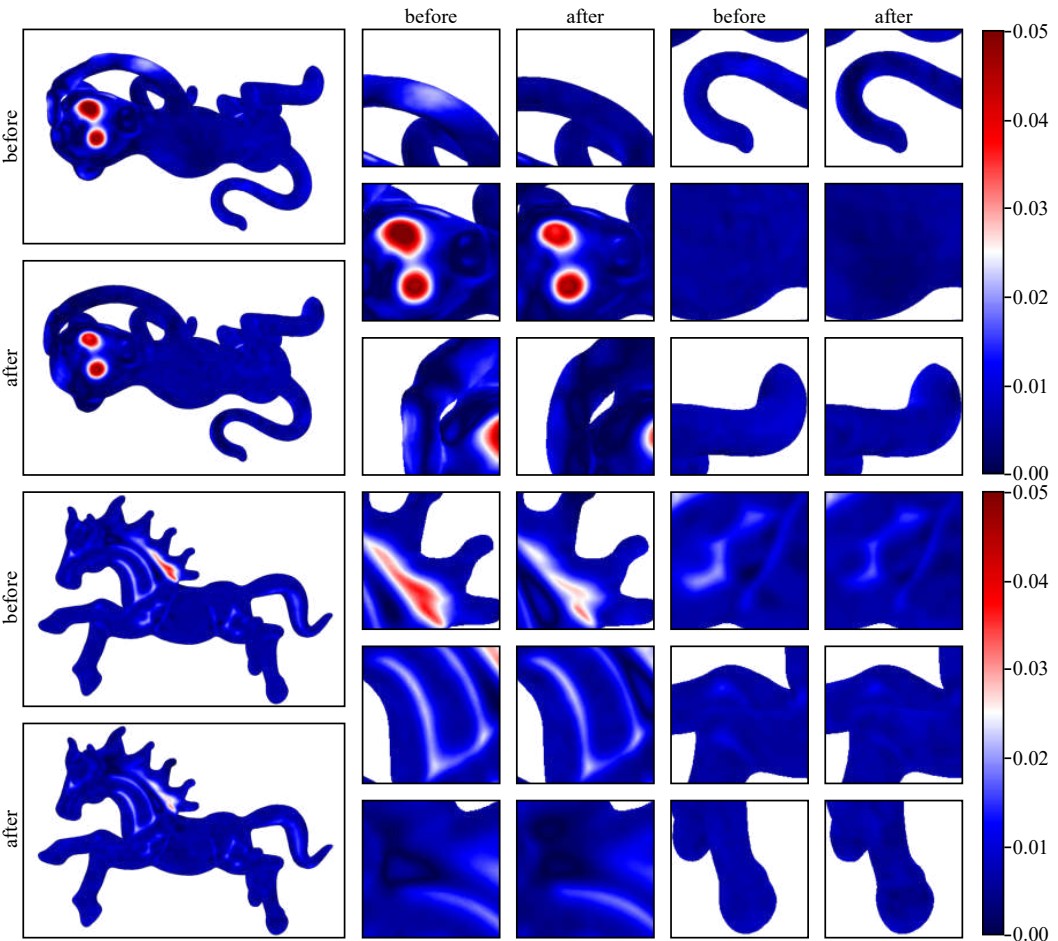

Figure 11: Per-vertex heatmap illustrating the ground truth difference in Hausdorff Distance before and after optimization.

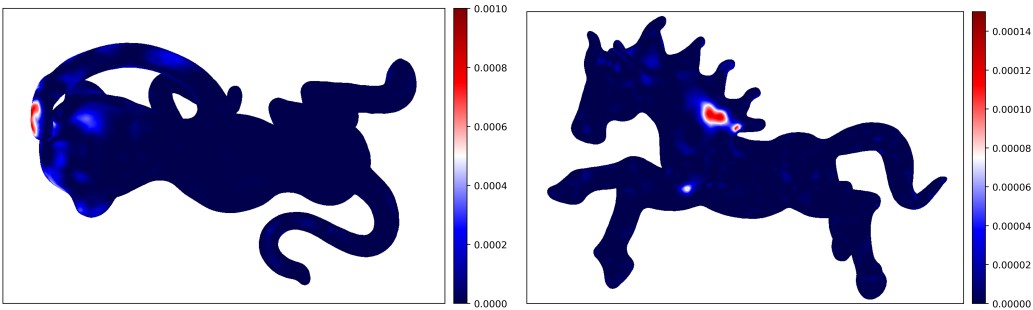

Figure 12: Per-Vertex Chamfer Distances betweeen the meshes before and after optimization.

### A.6.4 ABLATION ON DIFFERENT IORS

We conducted experiments by using different index of refraction (IoR) settings for the same object and performing reconstruction. Table 9 shows the recovered IoR values, which are close to the ground truth, demonstrating the stability of our method across different IoR settings. The results of novel view synthesis are presented in Figure 14.

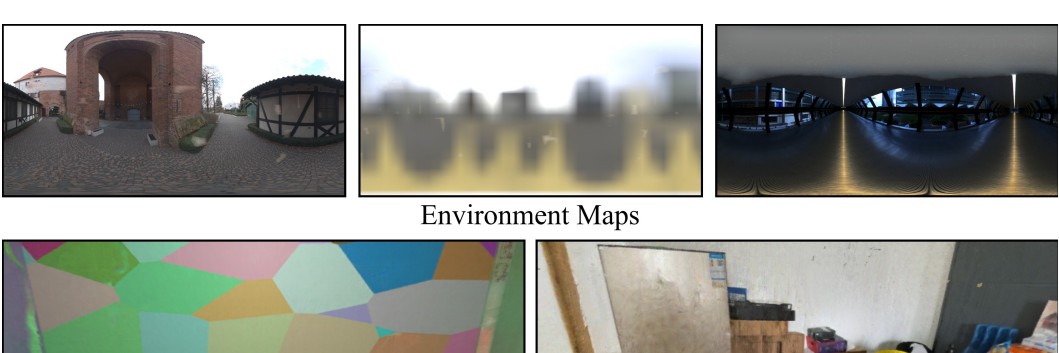

Environment Maps

Scene Captures

Figure 13: Visualization of the decomposed scene representations. Top: Recovered 2D environment maps (equirectangular projections) representing the far-field background. Bottom: 3D reconstructions of the scenes. The grey regions visible in the top-right environment map correspond to areas (zenith and nadir) that were not observed in the input footage due to the limited vertical field-of-view of the camera trajectory.

$$\eta = 1.3 \qquad \eta = 1.4 \qquad \eta = 1.5$$

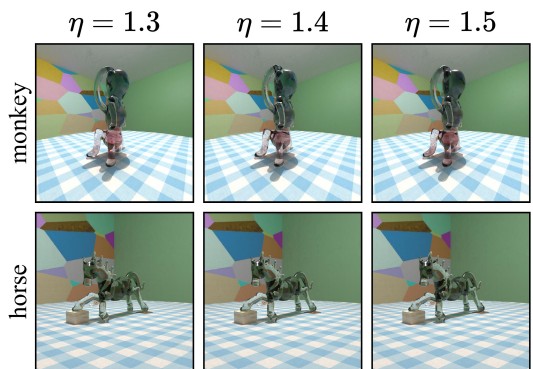

Figure 14: Rendering of the same object under different refractive indices (IOR).

## A.7 THE USE OF LARGE LANGUAGE MODELS

In the process of writing this paper, Large Language Models (LLMs) were not used. Both the research ideation and writing were fully carried out by the authors.

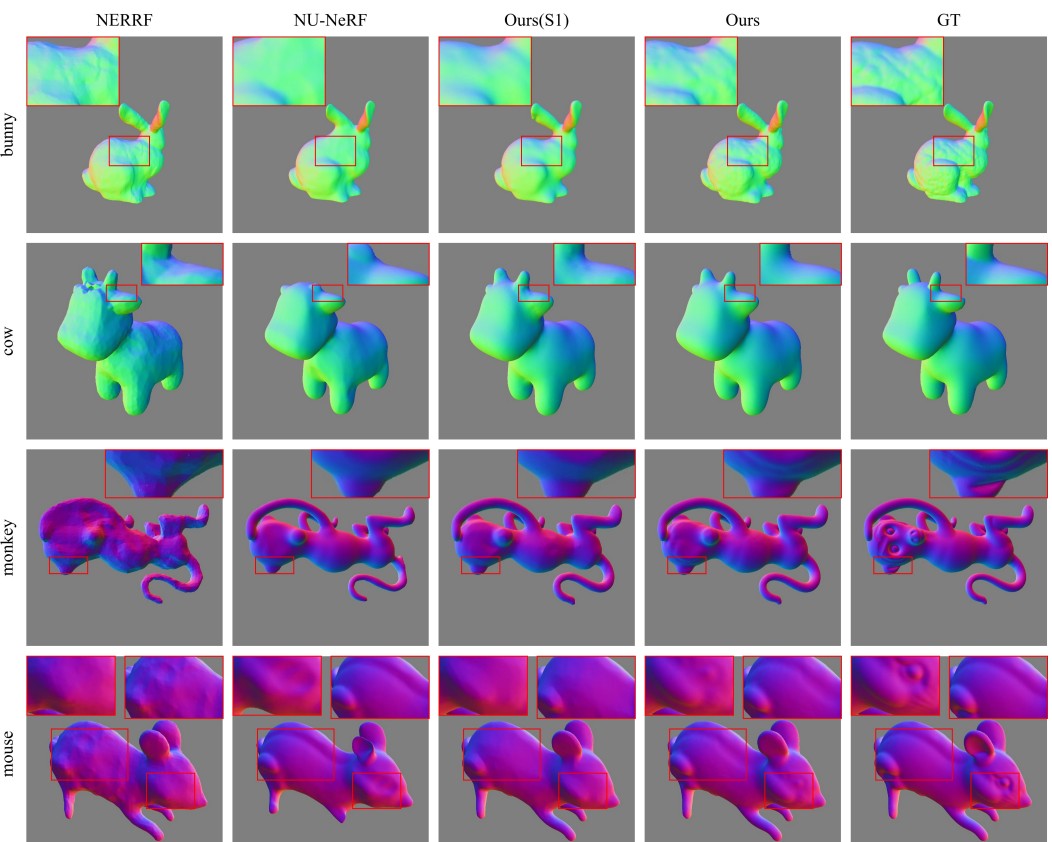

Figure 15: Normal maps comparison. The Ours(S1) column shows the normal maps rendered from the first stage of our method.

| Method | NeRO | NU-NeRF | NeRRF | Ours(S1) | Ours(S1)* | Ours(S1)** | Ours | Ours* | Ours** |
|---|---|---|---|---|---|---|---|---|---|
| CD($\times 10^{-4}$)↓ | 36.022 | 7.891 | 13.341 | 4.666 | 4.329 | 4.468 | 3.264 | **3.203** | 3.341 |
| $F_1$($\times 10^{-1}$)↑ | 5.691 | 8.026 | 6.916 | 8.088 | 7.832 | 7.646 | 8.386 | **8.623** | 8.401 |

Table 8: The extended result of Table 1 in the paper. ∗ refers to the result obtained using noised masks and ∗∗ refers to the result obtained by cropping patches off the mask.

| Object | $\eta = 1.3$ | $\eta = 1.4$ | $\eta = 1.5$ |
|---|---|---|---|
| monkey | 1.288 | 1.381 | 1.473 |
| horse | 1.305 | 1.421 | 1.512 |

Table 9: Ablation study of IOR prediction under different ground-truths.

| Dataset | CD($\times 10^{-3}$)↓ | | | | NC($\times 10^{-1}$) | | | |
|---|---|---|---|---|---|---|---|---|
| Method | horse | monkey | frame | block | horse | monkey | frame | block |
| only $\mathcal{L}_{\text{geo-init}}$ | 3.233 | 4.915 | 1.652 | 6.969 | 2.403 | 2.498 | 2.431 | 2.794 |
| + $\mathcal{L}_{\text{dilation}}$ | 0.481 | 1.039 | 14.683 | 31.602 | 1.298 | 1.713 | 0.747 | 0.933 |
| + $\mathcal{L}_{\text{smooth}}$ | 0.594 | 1.139 | 14.847 | 21.456 | 0.412 | 0.776 | 0.747 | 0.210 |
| full | 0.484 | 0.459 | **1.072** | 4.457 | 0.245 | 0.166 | 0.381 | 0.101 |
| after processing | **0.456** | **0.458** | 1.084 | **1.853** | **0.052** | **0.053** | **0.110** | **0.028** |

Table 10: Ablation study of initial geometry reconstruction. The second best results are underlined.

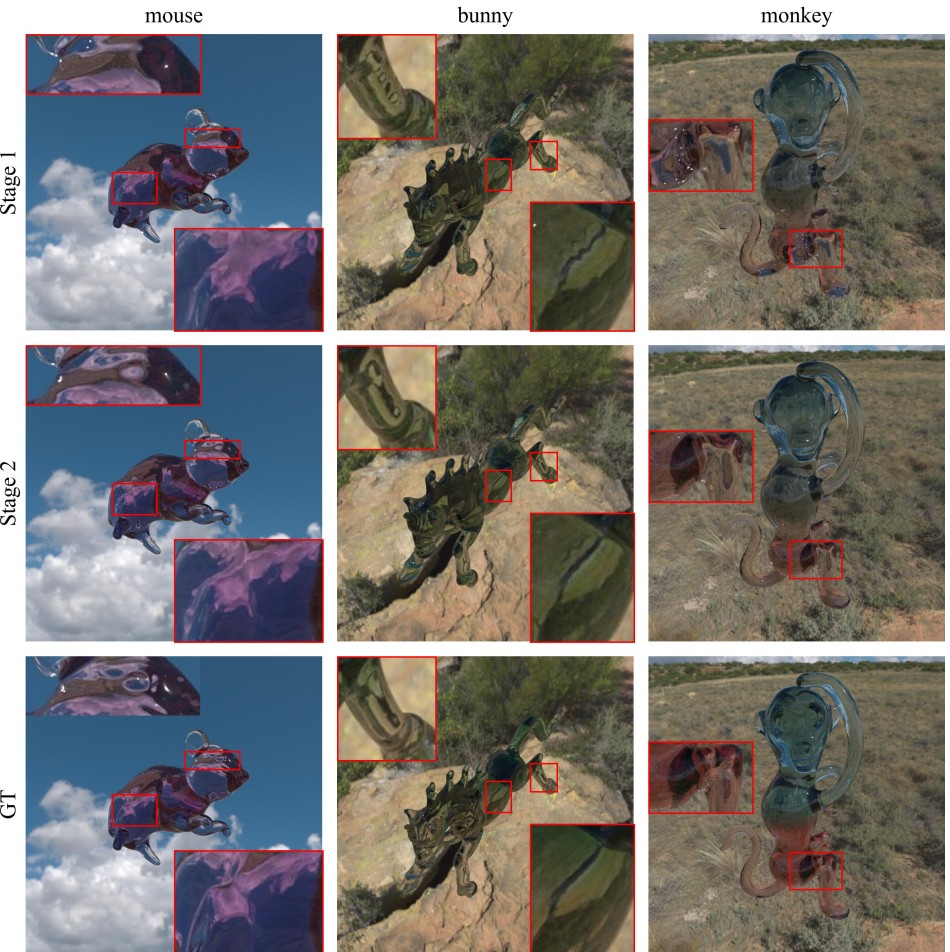

Figure 16: A comparison of relighting results using the coarse mesh from stage 1 and the refined mesh from stage 2.

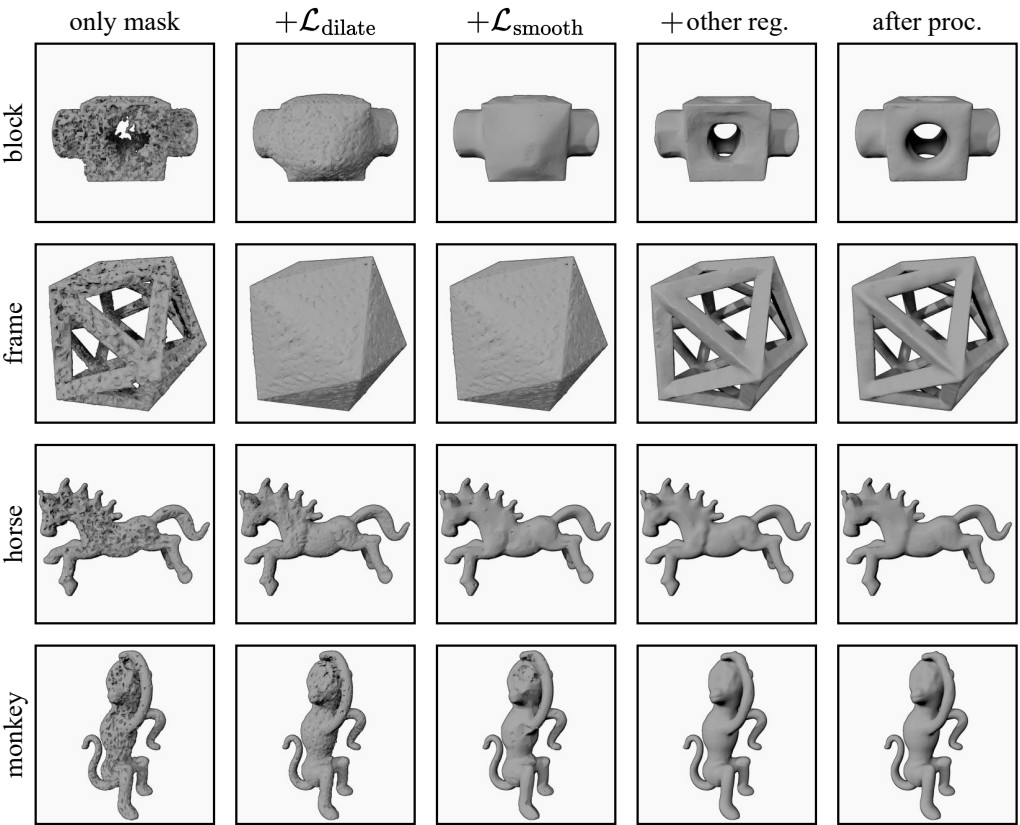

Figure 17: Ablation studies for different regularization.

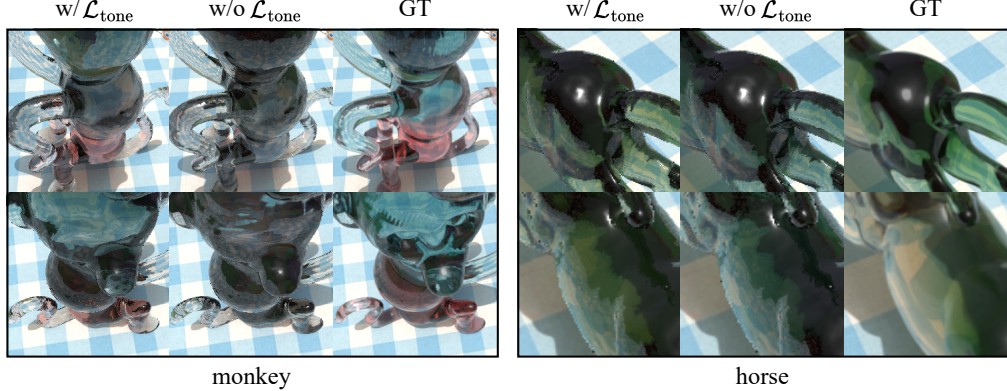

Figure 18: Reconstruction with/without the tone regularization.

