# OpenReview forum: "DiffTrans: Differentiable Geometry-Materials Decomposition for Reconstructing Transparent Objects"
_ICLR.cc/2026/Conference — ICLR 2026 Poster_

### Official Review · Reviewer_6MNx · 2025-10-19

**Soundness:** 3
**Presentation:** 4
**Contribution:** 3
**Rating:** 4
**Confidence:** 5

**Summary:**

This paper proposes DiffTrans, a differentiable rendering framework for reconstructing the 3D geometry and material properties (index of refraction and absorption rate) of transparent objects from multi-view images. The method operates in three stages: 1) Geometry Initialization: Using FlexiCubes and multi-view object masks to reconstruct a coarse mesh, regularized for smoothness and completeness. 2) Environment Initialization: Recovering the surrounding scene lighting using a radiance field from the out-of-mask image regions. 3) Joint Optimization: A novel, efficient, recursive differentiable ray tracer implemented in OptiX/CUDA that jointly refines the geometry, IoR, and absorption rate in an end-to-end manner. The key advantage is the ability to handle transparent objects with complex geometry, enabling high-quality reconstruction and relighting. Experimental results and ablation studies demonstrate nice results compared with the state-of-the-art methods.

**Strengths:**

Basically, the explicit and joint optimization of absorption rate alongside geometry and IoR is a significant step beyond prior work, which often focused only on surface properties or ideal transparency. This directly addresses a major limitation in reconstructing real-world objects.

Overall, the work combines several advanced techniques (FlexiCubes for mesh initialization, a radiance field for environment, and a custom differentiable ray tracer) into a cohesive, progressive pipeline. The integration of a mesh-based differentiable ray tracer, as opposed to a volume-based or implicit one, is a distinct choice for this problem.

The design of the recursive ray tracer and the specific regularization terms (like the tone loss L_tone) are novel contributions tailored to the ill-posed nature of reconstructing absorptive transparent objects.

**Weaknesses:**

First of all, reconstructing transparent objects has always been a very challenging problem in both the vision and graphics communities. The method proposed in this paper takes another step forward in this difficult area. Although the results are quite good, I still have several questions. I am open to hearing the authors’ feedback.

(1) In fact, this method can be categorized as a mesh optimization approach rather than a neural radiance one. Therefore, I am curious why it is not compared with [Lyu et al. 2020]. In other words, the comparison in Table 1 seems quite unfair, as all other methods are NeRF-based, which are inherently inferior in reconstruction quality compared to mesh optimization approaches.

(2) The initialization seems to have a significant impact. As shown in Table 1, the optimization process itself does not lead to a very noticeable improvement. I would like to see a comparative visualization, such as a per-vertex heatmap showing the difference in Chamfer Distance / Hausdorff Distance before and after optimization, which would be more intuitive to check the effectiveness of optimization.

(3) IOR optimization has always been a difficult problem. I suggest that the authors analyze the recovery of the IOR under different refractive indices for the same object. As the IOR increases, the proportion of total internal reflection rises sharply, making IOR estimation more challenging. It is unclear whether the ray tracer can correctly handle such cases.

(4) Regarding the environment map, it would be helpful to include some results showing the recovery of the environment map.

(5) The geometric details could be better illustrated using a normal map, since the current white model rendering makes it hard to perceive geometric enhancement. From Fig. 4, for example, the eyes of the Monkey model are almost not recovered at all. Another useful metric would be the MAE between the predicted and ground-truth outgoing ray directions.

(6) In recursive ray tracing, it is not mentioned how the method handles cases of internal total reflection.

(7) The training time for a single object? Any failure cases? Moreover, given that the optimization involves a large number of variables, I wonder whether the entire optimization process remains stable.

**Questions:**

As discussed above.

---

> ### Author Response · Authors · 2025-11-24
>
> Thank you for your insightful feedback and thorough review of our paper. We carefully respond to each of the concerns and questions below. Due to the word limit we split the reply into two comments.
>
> **[Q1]:** The comparison in Table 1 seems quite unfair, as all other methods are NeRF-based, which are inherently inferior in reconstruction quality compared to mesh optimization approaches.
>
> **[A1]:** Thanks for your suggestion. We would like to clarify that NeRRF uses the Deep Marching Tetrahedra (DMTet) representation, which is also a mesh optimization method. While NU-NeRF is a NeRF-based approach, it currently represents the state-of-the-art for transparent object reconstruction, with geometric reconstruction quality surpassing that of all previous mesh optimization methods. Therefore, we chose to compare our approach with these methods. On the other hand, mesh optimization methods such as DRT [1] and NeTO [2] rely on very strong priors and assumptions, such as requiring specialized capture devices to obtain outgoing rays or hit positions on the background, which makes them difficult to apply to general in-the-wild scenarios. In contrast, our method only requires a monocular video with known camera poses to achieve reconstruction, making it far more practical for real-world applications.
>
> **[Q2]:** A comparative visualization, such as a per-vertex heatmap showing the difference in Chamfer Distance / Hausdorff Distance before and after optimization.
>
> **[A2]:** Thanks for your suggestion. We have provided the rendered heatmap of the differences between the Stage 1 and the Stage 2 meshes in Figure 11 and Figure 12 of the appendix.
>
> **[Q3]:** Analyze the recovery of the IoR under different refractive indices for the same object.
>
> **[A3]:** We conducted experiments by using different index of refraction (IoR) settings for the same object and performing reconstruction. The table below shows the recovered IoR values, which are close to the ground truth, demonstrating the stability of our method across different IoR settings. The results of novel view synthesis are presented in Figure 14 of the appendix.
>
> | Object | $\eta=1.3$ | $\eta=1.4$ | $\eta=1.5$ |
> | ------ | ---------- | ---------- | ---------- |
> | monkey | 1.288      | 1.381      | 1.473      |
> | horse  | 1.305      | 1.421      | 1.512      |
>
> **[Q4]:** It would be helpful to include some results showing the recovery of the environment map.
>
> **[A4]:** Thanks. We have attached some visualization of the recovery of the environment map in Figure 13 of the appendix.
>
> **[Q5]:** The geometric details could be better illustrated using a normal map, since the current white model rendering makes it hard to perceive geometric enhancement.
>
> **[A5]:** We visualize the normal maps of the reconstructed geometry for the *bunny*, *cow*, *monkey* and *mouse* models in Figure 15 of the appendix. Compared to the baseline methods, the mesh reconstructed by our DiffTrans exhibits geometry details that are closer to the ground truth and is also smoother.
>
> **[Q6]:** In recursive ray tracing, it is not mentioned how the method handles cases of internal total reflection.
>
> **[A6]:** Thanks. Total internal reflection is handled in a physically consistent manner: once the conditions for total internal reflection are satisfied, we generate a reflected ray and propagate it through the ray tree as a perfect reflection.
>
> **[Q7]:** The training time for a single object?
>
> **[A7]:** Thanks. All experiments were conducted on NVIDIA RTX 3090 GPUs with 24 GB of VRAM. The training time for our scenes fell within 1–2 hours, depending on the geometric complexity of the object. Higher geometric complexity requires more ray bounces for a fixed batch size, which in turn increases the per-iteration training time.

---

> ### Author Response · Authors · 2025-11-24
>
> **[Q8]:** Any failure cases?
>
> **[A8]:** Due to the fact that even minor inaccuracies in geometry or in the estimated IoR can cause substantial deviations in the outgoing ray direction and position, our method exhibits reduced reliability in highly concave regions, for instance, around the monkey’s eye sockets in Figure 4 of the main paper. Nonetheless, baseline methods demonstrate similarly limited performance in this scenario, whereas our approach still achieves markedly superior reconstruction quality overall.
>
> **[Q9]:** Given that the optimization involves a large number of variables, I wonder whether the entire optimization process remains stable.
>
> **[A9]:** Although our method involves optimizing multiple explicit scene parameters, we observe that the optimization process remains stable. In fact, we employ the same set of hyperparameters across all scenes, and the results, reported in the table under entries marked with an asterisk (\*), achieve comparable levels of accuracy. This consistency further substantiates the robustness and stability of our approach.
>
> | Method                           | NeRO   | NU-NeRF | NeRRF  | Ours(S1) | Ours(S1)$^*$ | Ours      | Ours$^*$  |
> | -------------------------------- | ------ | ------- | ------ | -------- | ------------ | --------- | --------- |
> | CD($\times10^{-4}$)$\downarrow$  | 36.022 | 7.891   | 13.341 | 4.666    | 4.301        | **3.264** | 3.350     |
> | $F_1$($\times10^{-1}$)$\uparrow$ | 5.691  | 8.026   | 6.916  | 8.088    | 7.781        | 8.386     | **8.723** |
>
> [1] Differentiable Refraction-Tracing for Mesh Reconstruction of Transparent Objects.
>
> [2] NeTO: Neural Reconstruction of Transparent Objects with Self-Occlusion Aware Refraction-Tracing.

---

> > ### Comment · Reviewer_6MNx · 2025-11-27
> >
> > Thanks for the detailed rebuttal. The updated results resolve all my concerns and demonstrate the effectiveness of the proposed methods, I have no further issues and will raise my score toward acceptance.

---

> > > ### Author Response · Authors · 2025-11-28
> > >
> > > Thanks for your feedback and for raising your score. It’s incredibly motivating for us, and we’re deeply grateful for the effort and time you’ve dedicated to our paper.

---

### Official Review · Reviewer_RGa4 · 2025-10-27

**Soundness:** 3
**Presentation:** 2
**Contribution:** 3
**Rating:** 6
**Confidence:** 4

**Summary:**

This paper proposes DiffTrans, a framework for jointly reconstructing geometry and performing inverse rendering from multi-view images of transparent objects. It first initializes the geometry using the silhouette from the object mask with several regularization losses. Then, it refines the geometry and optimizes the refraction parameters using differentiable ray tracing. With the inverse rendering results, it supports applications such as relighting and demonstrates superior quality to baseline SOTA methods.

**Strengths:**

1. The proposed framework successfully solved the highly challenging transparent object reconstruction and inverse rendering task, while previous works struggled to handle them simultaneously.
2. The proposed differentiable rendering algorithm in Sec 3.3 is physically based and technically sound, following the physical laws of light transport.
3. The experimental results are significantly superior to baseline methods.

**Weaknesses:**

1. Related work and citations. This paper is an inverse and differentiable rendering paper, but Section 2 does not survey related work in these research directions. I suggest adding a paragraph for a comprehensive review of existing inverse/differentiable rendering and relighting methods.
2. Paper writing. A lot of necessary details and explanations are missing in the paper, making it hard to understand for people without a background in the related physics laws. Please refer to the "Questions" part for some specific confusions when I read the paper, and the authors should include these details/explanations in the revised paper.
3. Baseline selection. The paper selects NeRO as one of its baselines, but NeRO does not support refraction objects at all, leading to an unfair comparison. It would be better to choose baselines supporting transparent objects.
4. Geometry initialization. A comparison between the initialized geometry from the first stage and the refined geometry from the last stage should be included, so that we can know how differentiable rendering helps the geometry refinement.

Several example citations in differentiable and inverse rendering (both mesh-based and NeRF-based, including but not limited to):
```
@article{zhang2021nerfactor,
  title={Nerfactor: Neural factorization of shape and reflectance under an unknown illumination},
  author={Zhang, Xiuming and Srinivasan, Pratul P and Deng, Boyang and Debevec, Paul and Freeman, William T and Barron, Jonathan T},
  journal={ACM Transactions on Graphics (ToG)},
  volume={40},
  number={6},
  pages={1--18},
  year={2021},
  publisher={ACM New York, NY, USA}
}
@inproceedings{jin2023tensoir,
  title={Tensoir: Tensorial inverse rendering},
  author={Jin, Haian and Liu, Isabella and Xu, Peijia and Zhang, Xiaoshuai and Han, Songfang and Bi, Sai and Zhou, Xiaowei and Xu, Zexiang and Su, Hao},
  booktitle={Proceedings of the IEEE/CVF Conference on Computer Vision and Pattern Recognition},
  pages={165--174},
  year={2023}
}
@inproceedings{gao2024relightable,
  title={Relightable 3d gaussians: Realistic point cloud relighting with brdf decomposition and ray tracing},
  author={Gao, Jian and Gu, Chun and Lin, Youtian and Li, Zhihao and Zhu, Hao and Cao, Xun and Zhang, Li and Yao, Yao},
  booktitle={European Conference on Computer Vision},
  pages={73--89},
  year={2024},
  organization={Springer}
}
@inproceedings{dai2025inverse,
  title={Inverse Rendering using Multi-Bounce Path Tracing and Reservoir Sampling},
  author={Dai, Yuxin and Wang, Qi and Zhu, Jingsen and Xi, Dianbing and Huo, Yuchi and Qian, Chen and He, Ying},
  booktitle={The Thirteenth International Conference on Learning Representations},
  year = {2025},
  url = {https://openreview.net/forum?id=KEXoZxTwbr}
}
@article{son2024dmesh,
  title={DMesh: A Differentiable Mesh Representation},
  author={Son, Sanghyun and Gadelha, Matheus and Zhou, Yang and Xu, Zexiang and Lin, Ming C and Zhou, Yi},
  journal={arXiv preprint arXiv:2404.13445},
  year={2024}
}
@article{binninger2025tetweave,
  title={TetWeave: Isosurface Extraction using On-The-Fly Delaunay Tetrahedral Grids for Gradient-Based Mesh Optimization},
  author={Binninger, Alexandre and Wiersma, Ruben and Herholz, Philipp and Sorkine-Hornung, Olga},
  journal={ACM Transactions on Graphics (TOG)},
  volume={44},
  number={4},
  pages={1--19},
  year={2025},
  publisher={ACM New York, NY, USA}
}
```
I generally like the paper's proposed method, but the current writing quality needs significant improvement.

**Questions:**

1. In Eq. 1, what are the points $x_i$? Are they mesh vertices, points sampled from the surface of the mesh, or points sampled randomly in the space? Besides, `\mathcal` is missing for the $\mathcal L$ symbol.
2. What are the optimizable refraction material parameters in the last stage (Refine phase)? Are they the IoR $\eta$ and the absorption rate $\mu_t(\mathbf{x})$? Is $\eta$ a single scalar while $\mu_t(\mathbf{x})$ a spatially-varying field?
3. In Eqs. 5 and 6, what are $\omega^\parallel_t$ and $\omega^\perp_t$? I guess they refer to the "parallel and perpendicular part" of the direction, but I don't know their specific physical meanings, and I think this should be explained.
4. For the environment initialization, did you handle the LDR-HDR issue? Since the environment NeRF is learned from LDR images, the lighting queried in the ray tracing stage will remain LDR. Using LDR values to compute the rendering equation may lead to overdark estimation, and also be vulnerable to imaging artifacts such as over-/under-exposure.

---

> ### Author Response · Authors · 2025-11-24
>
> Thank you for your insightful feedback and thorough review of our paper. We carefully respond to each of the concerns and questions below.
>
> **[Q1]:** I suggest adding a paragraph for a comprehensive review of existing inverse/differentiable rendering and relighting methods.
>
> **[A1]:** Thanks for your suggestion! Existing inverse/differentiable rendering and relighting methods have provided many insights for our method. The references you kindly provided depicted a view of some important monuments of its development. We have added them in blue texts in the Section 2 of the main paper.
>
> **[Q2]:** Baseline selection. The paper selects NeRO as one of its baselines, but NeRO does not support refraction objects at all, leading to an unfair comparison.
>
> **[A2]:** Thanks for your suggestion. Similar to prior work, we compare against reflection-only methods such as NeRO to demonstrate the importance of modeling transparency effects. In addition, we include comparisons with refraction-aware approaches such as NeRRF and NU-NeRF to ensure a fair and comprehensive evaluation.
>
> **[Q3]:** A comparison between the initialized geometry from the first stage and the refined geometry from the last stage should be included.
>
> **[A3]:** Thanks for your suggestion. Besides the quantitative results in Table 1 of the main paper, we provided a rendered heatmap of the differences between the Stage 1 and the Stage 2 meshes in the Figure 11 and the Figure 12 of appendix.
>
> **[Q4]:** In Eq.1, what are the points $\mathbf{x}_i$? Are they mesh vertices, points sampled from the surface of the mesh, or points sampled randomly in the space?
>
> **[A4]:** Thanks for your comments. The points $\mathbf{x}_i$ are sampled randomly in the space.
>
> **[Q5]:** What are the optimizable refraction material parameters in the last stage (Refine phase)? Are they the IoR $\eta$ and the absorption rate $\mu_t(\mathbf{x})$? Is $\eta$ a single scalar while $\mu_t(\mathbf{x})$ a spatially-varying field?
>
> **[A5]:** In the last stage, the mesh vertices, inner absorptive rate and IoR is jointly optimized. And $\eta$ a single scalar while $\mu_t(\mathbf{x})$ a spatially-varying field.
>
> **[Q6]:** In Equ. 5 and 6, what are $\omega_t^\perp$ and $\omega_t^\parallel$?
>
> **[A6]:** $\omega_t^\perp$ and $\omega_t^\parallel$ are two components of $\omega_t$ perpendicular and parallel to the surface normal $\mathbf{n}$, respectively. From Equ. 4 to Equ. 6, we are computing the direction of an out-going refracted light using Snell's law. Take $\theta_i$ and $\theta_t$ as the angle between $\mathbf{n}$ of $\omega_i$ and $\omega_t$, we have $\sin\theta_i=\eta\sin\theta_t$. Then by decomposing $\omega_i$ into the vector in direction of $\mathbf{n}$ and another parpendicular component and algebraic calculations, we can obtain the directional vector of the $\omega_t$.
>
> **[Q7]:** For the environment initialization, did you handle the LDR-HDR issue? Since the environment NeRF is learned from LDR images, the lighting queried in the ray tracing stage will remain LDR. Using LDR values to compute the rendering equation may lead to overdark estimation, and also be vulnerable to imaging artifacts such as over-/under-exposure.
>
> **[A7]:** Thanks for your comments. For synthetic scenes, we use HDR environments. For real-world captures with LDR inputs, we apply gamma correction with $\gamma=2.2$ to mitigate the domain gap between LDR input images and HDR environment maps. In Figure 13 of the appendix, we provide visualizations of the recovered environment maps along with the reconstructed 3D scenes, which demonstrate that the overall quality is satisfactory.

---

> > ### Comment · Reviewer_RGa4 · 2025-11-25
> > **Further discussion about geometry refinement**
> >
> > Thank you for your detailed response. Most of my major concerns have been addressed. Nevertheless, I'd like to point out 2 minor remaining issues:
> > 1. Fig. 9 in the Appendix looks broken on MacBook's Safari browser (although it looks normal on Google Chrome). Maybe there is a PDF encoding/compilation issue of the submission, and I suggest a double-check.
> > 2. In Fig. 11, I can indeed observe the improvements of the geometry before and after refinement, but the improvement looks marginal. I'm actually surprised that the initial geometry already looks quite reasonable with only the supervision from the mask and several regularizations. I agree with Reviewer 6MNx that geometry details may be better illustrated by normal maps, especially for rendering, since normal directions determine the refraction direction and even slight normal perturbations may cause significant rendering changes. Therefore, it would be great if the authors could provide qualitative comparisons of the normal map before and after the refinement, and if possible, compare the rendering results using the geometry before and after the refinement.

---

> > > ### Comment · Reviewer_wurg · 2025-11-26
> > >
> > > Although I am only a reviewer of this paper, I'd like to explain why S1 can generate a good mesh. For FlexibleCube, reconstructing a convex object from a mask is relatively straightforward, as multi-view reconstruction based solely on silhouettes is not particularly challenging. The main difficulty for such methods lies in capturing concave structures that are not visible in the silhouette. For example, the eyes in the "monkey" model cannot be well captured from any camera viewpoint, leading to higher geometric errors, as shown in Figure 11.

---

> > > ### Author Response · Authors · 2025-11-26
> > >
> > > **[Q8]:** Broken PDF on Macs.
> > >
> > > **[A8]:** Thank you for your response. We have discovered that Figure 9 in the appendix was indeed corrupted. We have since repaired the issue and re-uploaded the corrected version.
> > >
> > > **[Q9]:** Normal map between the meshes in the two stages and corresponding rendering result.
> > >
> > > **[A9]:** Thanks for your valuable suggestion. We have included a comparison of normal maps in Figure 15 of the appendix. Regarding the rendering results, since stage 1 only reconstructs the geometry of the object, we utilized the materials recovered in stage 2 for rendering. The results are presented in Figure 16 of the appendix.

---

> > > > ### Comment · Reviewer_RGa4 · 2025-11-26
> > > > **Issues addressed**
> > > >
> > > > Thank you for your response and the updated results. I think my issues are mostly addressed, and the additional results of the normal map and rendering comparison effectively validates the geometry refinement, making the paper’s evaluation more complete. I have no further issues and will raise my score toward acceptance.

---

> > > > > ### Author Response · Authors · 2025-11-28
> > > > >
> > > > > Thank you for your thoughtful response and for raising your score. This is a great encouragement for us, and we truly appreciate the time and effort you put into reviewing our work.

---

### Official Review · Reviewer_wurg · 2025-11-01

**Soundness:** 3
**Presentation:** 2
**Contribution:** 3
**Rating:** 4
**Confidence:** 4

**Summary:**

This paper presents a novel multi-view image-based 3D reconstruction framework tailored for transparent objects. By adopting a two-stage design, the method achieves a more accurate geometric representation.

In Stage 1, the authors model an initial outer surface of the transparent object using object masks. To ensure stable convergence, several operations—including dilation—and regularization terms are introduced to constrain the mesh shape. Additionally, non-mask regions are leveraged to learn an environment field that captures scene appearance and lighting.

In Stage 2, a custom optical model is employed for joint optimization, enabling simultaneous recovery of the object’s material properties and refinement of the geometry and appearance from Stage 1.

Experimental results demonstrate that the proposed method achieves state-of-the-art (SOTA) performance.

The main contributions of this work are:

1) A two-stage reconstruction pipeline that produces a high-quality initialization robust to the challenges of transparency.

2) A novel rendering pipeline capable of simulating light transport through transparent materials.

3) A new real-world dataset specifically captured for transparent object reconstruction.

**Strengths:**

1) This paper implements a ray tracer for transparent materials based on OptiX and CUDA, which serves as a valuable contribution to the community.

2) To obtain a high-quality initial mesh suitable for ray tracing, the paper proposes a flexible cube-based modeling approach.

**Weaknesses:**

1) The amount of real-world test data is insufficient—only a single real captured object is used for evaluation. Moreover, the rendered results exhibit inaccurate material appearance, and the paper does not provide any geometric visualization (e.g., mesh or point cloud) of this real object. This raises concerns about the method’s practicality. In contrast, methods like NU-NeRF validate their approach on multiple real transparent objects, offering stronger empirical support.

2) The proposed method relies heavily on an accurate foreground mask for Stage 1 training. In the paper, masks for real data are obtained by combining SAM with manual refinement, which undermines the method’s applicability in fully automatic settings. Notably, prior work such as [1] also leverages vision models for transparent object reconstruction but avoids human intervention entirely, making it more scalable and practical. Also, you may cite this paper.


[1] TSGS: Improving Gaussian Splatting for Transparent Surface Reconstruction via Normal and De-lighting Priors

**Questions:**

1) In the ablation study in Figure 10, does “other reg” refer to the original regularization losses used in FlexibleCube?

2) How does the proposed method perform on the less obvious transparent objects in the NU-NeRF dataset? Would the presence of opaque objects inside transparent ones in the NU-NeRF dataset negatively impact—or even cause failure in—the reconstruction using the proposed method? The approach appears to be designed primarily for solid transparent objects, and may not handle internal occluders or complex layered transparency effectively.

3) Many structural details of transparent objects cannot be captured by silhouette masks alone. Although the paper suggests that the two-stage optimization can recover such missing structures, could the authors provide a direct geometric comparison between the results of Stage 1 and Stage 2 to demonstrate this improvement?

4) Could the authors provide a full 360-degree video rendering of the results?

---

> ### Author Response · Authors · 2025-11-24
>
> Thank you for your insightful feedback and thorough review of our paper. We carefully respond to each of the concerns and questions below.
>
> **[Q1]:** The amount of real-world test data is insufficient.
>
> **[A1]:** Thanks for your suggestion. We have provided more results on real-world scenes in Figure 8 of the appendix.
>
> **[Q2]:** The proposed method relies heavily on an accurate foreground mask for Stage 1 training.
>
> **[A2]:** Thanks for your comments. To evaluate the sensitivity of our method to mask accuracy, we introduced perturbations by randomly dilating and eroding the segmentation masks for each scene. The results, reported in the table under entries marked with an asterisk (\*), showed that the performance is largely unaffected, with metrics varying only within a small range. This demonstrates the robustness of our method with respect to mask accuracy.
>
> | Method                           | NeRO   | NU-NeRF | NeRRF  | Ours(S1) | Ours(S1)$^*$ | Ours         | Ours$^*$  |
> | -------------------------------- | ------ | ------- | ------ | -------- | ------------ | ------------ | --------- |
> | CD($\times10^{-4}$)$\downarrow$  | 36.022 | 7.891   | 13.341 | 4.666    | 4.329        | 3.264 | **3.203** |
> | $F_1$($\times10^{-1}$)$\uparrow$ | 5.691  | 8.026   | 6.916  | 8.088    | 7.832        | 8.386 | **8.623** |
>
> **[Q3]:** In the ablation study in Figure 10, does “other reg” refer to the original regularization losses used in FlexiCube?
>
> **[A3]:** Yes. "other reg" includes original regularizations used in FlexiCube.
>
> **[Q4]:** How does the proposed method perform on the less obvious transparent objects in the NU-NeRF dataset?
>
> **[A4]:** Thank you for your comment. NU-NeRF addresses a different problem setting: by predicting refracted rays using an implicit field, it can successfully reconstruct transparent objects enclosing nested opaque geometry. However, this formulation cannot handle transparent objects with complex materials, such as those exhibiting internal absorption rate, which is precisely the focus of our work. We acknowledge that our current method cannot be directly applied to nested scenarios. In future work, we plan to integrate implicit field representations with our differentiable recursive mesh ray tracer to address this limitation.
>
> **[Q5]:** Could the authors provide a direct geometric comparison between the results of Stage 1 and Stage 2 to demonstrate this improvement?
>
> **[A5]:** Yes. We have provided a rendered heatmap of the differences between the Stage 1 and the Stage 2 meshes in Figure 11 and Figure 12 of the appendix.
>
> **[Q6]:** Could the authors provide a full 360-degree video rendering of the results?
>
> **[A6]:** Thanks for your suggestion. We have included the videos in the supplementary materials.

---

> > ### Comment · Reviewer_wurg · 2025-11-25
> >
> > Thank you for addressing my concerns. Considering that this paper presents a novel approach for transparent object reconstruction and provides a differentiable codebase that is valuable to the community, I have decided to raise my score. However, the diversity of real-world data remains quite limited. The newly added one data is still very similar to the original dataset. Therefore, the real-world generalization capability of the proposed method is still questionable, and I adjust my final score to 6.

---

> > ### Comment · Reviewer_wurg · 2025-11-26
> >
> > For A2, have you updated it in the PDF? Meanwhile, how do you randomly dilate and erode the segmentation masks? If you only apply a single 3×3 dilation, it should be sufficient for FlexibleCube to reconstruct the model. In fact, most mask prediction errors manifest as localized block-like missing regions, rather than simple erosion or dilation.

---

> > > ### Author Response · Authors · 2025-11-26
> > >
> > > Thank you sincerely for your kind follow-up and for raising your rating. We are glad our rebuttal has addressed your concerns, and we appreciate your constructive review. Below is our further reply:
> > >
> > > We have updated **[A2]** in the appendix. Regarding the perturbation of the masks, we randomly applied multiple 3$\times$3 dilations and erosions to different regions of the segmentation masks. To further assess the robustness of our method, we conducted additional experiments. For each scene, we randomly selected 70% of the camera viewpoints and introduced perturbations by removing square regions of 10 to 20 pixels in length at 1 to 3 random positions within the mask. The results of these experiments, presented in Table 8 of the appendix, show that the metrics fluctuate within a narrow range, with minimal impact on the overall performance.

---

### Official Review · Reviewer_jaxD · 2025-11-02

**Soundness:** 3
**Presentation:** 3
**Contribution:** 3
**Rating:** 6
**Confidence:** 3

**Summary:**

This paper presents DiffTrans, a differentiable rendering framework for reconstructing transparent objects with complex geometry and internal texture from multi-view images. It jointly optimizes geometry and material properties—specifically the index of refraction (IoR) and absorption rate—in an end-to-end manner. The method initializes geometry using FlexiCubes and estimates illumination via an environment radiance field, then refines both using a recursive differentiable ray tracer implemented in CUDA/OptiX. Experiments on synthetic and real data demonstrate strong reconstruction and relighting performance.

**Strengths:**

The work tackles a highly challenging problem in differentiable rendering, enabling reconstruction and relighting of transparent objects with complex internal geometry and refractions. The decoupling of geometry and materials of transparent objects also presents a reasonable path for modeling light refraction and absorption within an end-to-end differentiable pipeline. The method is efficiently implemented in CUDA/OptiX with strong empirical performance on both synthetic and real-world data.

**Weaknesses:**

The framework relies on several simplifying assumptions, uniform refractive index, purely specular surfaces, and absorption-only materials, which could limit its applicability to more realistic transparent objects such as frosted, layered, or scattering materials.


The evaluation on real-world data is limited to a single “glass flower” scene, which does not sufficiently demonstrate robustness to real capture conditions such as noise, imperfect masks, or lighting variations. More diverse real-world examples and quantitative comparisons would strengthen the experimental section.

Although the recursive differentiable ray tracer is central to the method, its computational and memory efficiency are not analyzed in detail. Providing runtime comparisons or scalability analysis would help clarify its practical feasibility.

The method’s sensitivity to initialization (e.g., mask accuracy, environment estimation quality) is not thoroughly discussed. An ablation on initialization errors or noisy inputs would clarify robustness in real applications.

The authors could better articulate their conceptual distinction from prior works such as NeRRF, Nu-NeRF, and TransparentGS, to highlight their unique contributions more clearly. The authors should also discuss the conceptual differences from [1], which also highlights decoupling geometry and radiance for reconstructing semi-transparent surfaces, and may provide quantitative comparisons.

[1] Wu, Tianhao, et al. "αsurf: Implicit surface reconstruction for semi-transparent and thin objects with decoupled geometry and opacity." 2025 International Conference on 3D Vision (3DV). IEEE, 2025.

**Questions:**

Please refer to the above section for questions

---

> ### Author Response · Authors · 2025-11-24
>
> Thank you for your insightful feedback and thorough review of our paper. We carefully respond to each of the concerns and questions below.
>
> **[Q1]:** The framework relies on several simplifying assumptions.
>
> **[A1]:** Thanks for your comments. Inverse rendering is inherently an ill-posed problem, and the challenge is further amplified in the presence of transparent objects. Existing approaches typically rely on strong simplifying assumptions, such as assuming a known index of refraction or neglecting the material properties of transparency. In contrast, our method operates under more relaxed assumptions. To the best of our knowledge, it is the first approach capable of reconstructing transparent objects with spatially-varying absorption rate. Addressing more complex cases, such as frosted, layered, or scattering materials, remains an important direction that we plan to explore in future work.
>
> **[Q2]:** More diverse real-world examples.
>
> **[A2]:** Thanks for your suggestion. We have provided more results on real-world scenes in Figure 8 of the appendix.
>
> **[Q3]:** Computational and memory efficiency are not analyzed in detail.
>
> **[A3]:** Thank you for your comments. All experiments were conducted on NVIDIA RTX 3090 GPUs with 24 GB of VRAM. The training time for our scenes fell within 1–2 hours, depending on the geometric complexity of the object. Higher geometric complexity requires more ray bounces for a fixed batch size, which in turn increases the per-iteration training time. In our implementation, rays with more than four bounces are discarded during training. For inference, this threshold is increased to eight to achieve better visual quality. The memory consumption is approximately 20 GB during training with a batch size of 5,000 rays, and it never exceeded the available VRAM even in our most complex scene.
>
> **[Q4]:** The method’s sensitivity to initialization (e.g., mask accuracy, environment estimation quality) is not thoroughly discussed.
>
> **[A4]:** Thank you for your valuable comments. Since our method is based on ray tracing, the quality of the estimated environment can have a significant impact on the final reconstruction. However, we observed that the environment is relatively straightforward to reconstruct, even in real-world captured data. As shown in Figure 13 of the appendix, the reconstructed environments are of generally acceptable quality. In comparison, masks are more susceptible to various sources of noise. To evaluate the sensitivity of our method to mask accuracy, we introduced perturbations by randomly dilating and eroding the segmentation masks for each scene. The results, reported in the table under entries marked with an asterisk (*), show that the performance is largely unaffected, with metrics varying only within a small range. This demonstrates the robustness of our method with respect to mask accuracy.
>
> | Method                           | NeRO   | NU-NeRF | NeRRF  | Ours(S1) | Ours(S1)$^*$ | Ours         | Ours$^*$  |
> | -------------------------------- | ------ | ------- | ------ | -------- | ------------ | ------------ | --------- |
> | CD($\times10^{-4}$)$\downarrow$  | 36.022 | 7.891   | 13.341 | 4.666    | 4.329        | 3.264 | **3.203** |
> | $F_1$($\times10^{-1}$)$\uparrow$ | 5.691  | 8.026   | 6.916  | 8.088    | 7.832        | 8.386 | **8.623** |
>
> **[Q5]:** Conceptual distinction from prior works such as NeRRF, Nu-NeRF, TransparentGS and $\alpha$Surf.
>
> **[A5]:** Thanks for your comments. The primary distinction of DiffTrans from existing methods lies in its more relaxed assumptions, enabling the reconstruction of transparent objects with substantially more complex materials. NeRRF focuses on fully transparent objects and does not account for the materials. NU-NeRF only models the surface materials of transparent objects, while ignoring internal absorption rate. TransparentGS addresses a problem setting similar to NU-NeRF and adopts the 3D Gaussian Splatting representation. $\alpha$Surf considers only objects with semi-transparent surfaces and likewise does not model the volumetric materials. In contrast, our method targets transparent objects with more complex materials, jointly reconstructing both the IoR and the spatially-varying absorption rate, which fundamentally differentiates our approach from all prior work.

---

### Author Response · Authors · 2025-12-02

Dear PCs, SACs, ACs, and Reviewers,

We sincerely thank the reviewers for their valuable and constructive feedback. We are greatly encouraged to receive positive scores from all reviewers after the rebuttal phase, with the overall scores improving from the initial **6, 4, 6, 4** to **6, 6, 8, 6**. Specifically, reviewer wurg raised the score from 4 to 6, reviewer RGa4 from 6 to 8, and reviewer 6MNx from 4 to 6. Importantly, **all of these score increases occurred prior to the information-leak incident**.

All reviewers fully acknowledged the novelty and contributions of our approach. Reviewer jaxD noted that "**the work tackles a highly challenging problem**". Reviewer wurg characterized our approach as "**a novel multi-view image-based 3D reconstruction framework**" and "**a valuable contribution to the community**". Reviewer RGa4 remarked that our proposed framework "**successfully solved the highly challenging transparent object reconstruction and inverse rendering task**". Reviewers 6MNx recognized that our approach represents "**a significant step beyond prior work**" and provides "**novel contributions tailored to the ill-posed nature of reconstructing absorptive transparent objects**". The reviewers' questions primarily focused on the **experimental section**. In response to each reviewer’s questions, we have supplemented the following experiments:

- **The method’s stability, and sensitivity to initialization (Reviewers jaxD, wurg and 6MNx):** In Figure 13 of the appendix, we visualize the recovered environments, demonstrating that our method is capable of consistently reconstructing high-quality environments in both synthetic and real-world scenes. Additionally, in Table 8 of the appendix, we present reconstruction results under two different types of perturbations and noise applied to the segmentation masks, which further validate the robustness and stability of our method with respect to mask accuracy.
- **Direct geometric comparison between the results of Stage 1 and Stage 2 (Reviewers wurg, RGa4 and 6MNx):** We have provided a rendered heatmap of the differences between the Stage 1 and the Stage 2 meshes in Figure 11 and Figure 12 of the appendix. And we have included a comparison of normal maps in Figure 15, and a comparison of the rendering results in Figure 16.
- **More real-world test data (Reviewers jaxD and wurg):** We have provided the results on additional real-world scenes in Figure 8 of the appendix.
- **Analyze the recovery of the IoR under different refractive indices (Reviewer 6MNx):** We provide the recovered IoR values for the same scene under different IoR settings in the Table of **[A3]**. The recovered values are all close to the ground truth, demonstrating the stability of our method across varying IoR settings. The results of novel view synthesis are presented in Figure 14 of the appendix.
- **Others:** We provided a description of the training time and memory requirements (jaxD[A3] and 6MNx[A7]), a full 360-degree video of the results (wurg[A6]) and the recovered environment map (6MNx[A4]).

We provide our responses to the reviewers’ comments regarding the methodology and manuscript text as follows:

- **The framework relies on several simplifying assumptions (Reviewer jaxD):** Inverse rendering is inherently an ill-posed problem, and the challenge is further amplified in the presence of transparent objects. Existing approaches typically rely on strong simplifying assumptions, such as assuming a known index of refraction or neglecting the material properties of transparency. In contrast, our method operates under more relaxed assumptions. To the best of our knowledge, it is the first approach capable of reconstructing transparent objects with spatially-varying absorption rate.
- **Performance on the less obvious transparent objects in the NU-NeRF dataset (Reviewer wurg):** NU-NeRF addresses a different problem setting: by predicting refracted rays using an implicit field, it can successfully reconstruct transparent objects enclosing nested opaque geometry. However, this formulation cannot handle transparent objects with complex materials, such as those exhibiting internal absorption rate, which is precisely the focus of our work. We acknowledge that our current method cannot be directly applied to nested scenarios. In future work, we plan to integrate implicit field representations with our differentiable recursive mesh ray tracer to address this limitation.
- **Others:** We provided responses for the conceptual distinctions from previous methods (jaxD[A5]), baseline selection (RGa4[A2] and 6MNx[A1]), the LDR-HDR issue (RGa4[A7]), handling of internal total reflection (6MNx[A6]) and failure cases (6MNx[A8]).

We are glad that most of the reviewers’ concerns have been addressed. These constructive suggestions have made our work more comprehensive. We sincerely appreciate the time and effort from all reviewers.

Best regards,

Authors of Submission 4439

---

### Meta-Review · Area_Chair_PLf1 · 2025-12-24

**Summary:**

1) Insufficient real-world test data
2) The method relies heavily on the foreground mask
3) Better related work and paper writing, and illustrations requested by reviewers
4) Missing initialization analysis

**Reviewer Concerns:**

All reviewers were positive about the paper after the rebuttal and posted their opinion on the discussion board.

Reviewer wurg posted that their concern about real-world data is partially addressed. The authors add a new dataset during the rebuttal, but the reviewer expressed that this is still quite limited and he wished there were more data. Still, the reviewer decided to increase their score to a positive score, as evidenced by a posting on the discussion board.
Checking the additional work done during the rebuttal, I agree that there is still insufficient data and that this is a considerable weakness of the paper. Nevertheless, it is not a big enough weakness to warrant rejection.

There was also a longer discussion with reviewer RGa4 about technical details (initialization, rendering comparison, normal maps) and the reviewer concluded that "my issues are mostly addressed".

Reviewer 6MNx also posted that "the updated results resolve all my concerns and demonstrate the effectiveness of the proposed methods". The reviewer had multiple issues, also initialization, writing, and illustration of geometric details and acknowledges that the rebuttal addresses concerns.

**Reviewer Scores:**

The reviewers are expected to be either weakly accepting or accepting after the rebuttal. No reviewer is expected to have a negative opinion about the paper.

The initial reviews for the paper were:
jaxD: 6
wurg: 4 --> This reviewer commented on the discussion board that they acknowledged the rebuttal and that the score would be increased. I assume to a 6.
RGa4: 6 --> The reviewer acknowledged the rebuttal and discussed that the score would be increased. I assume to an 8.
6MNx: 4 --> The reviewer acknowledges the rebuttal and states the score would be raised. I assume to a 6. ("The updated results resolve all my concerns and demonstrate the effectiveness of the proposed methods. I have no further issues and will raise my score toward acceptance."

The final predicted scores would be 6, 6, 6, 8. The authors also claim in a posting that "the overall scores improving from the initial 6, 4, 6, 4 to 6, 6, 8, 6". This is credible and identical to my reading of the discussion board. I agree that all reviewers would be positive.

---

### Decision · Program_Chairs · 2026-01-26

Accept (Poster)